# The preeminence of ethnic diversity in scientific collaboration

Bedoor K. AlShebli [1], Talal Rahwan[1,2] & Wei Lee Woon [1,3]

Inspired by the social and economic benefits of diversity, we analyze over 9 million papers and 6 million scientists to study the relationship between research impact and five classes of diversity: ethnicity, discipline, gender, affiliation, and academic age. Using randomized baseline models, we establish the presence of homophily in ethnicity, gender and affiliation. We then study the effect of diversity on scientific impact, as reflected in citations. Remarkably, of the classes considered, ethnic diversity had the strongest correlation with scientific impact. To further isolate the effects of ethnic diversity, we used randomized baseline models and again found a clear link between diversity and impact. To further support these findings, we use coarsened exact matching to compare the scientific impact of ethnically diverse papers and scientists with closely-matched control groups. Here, we find that ethnic diversity resulted in an impact gain of 10.63% for papers, and 47.67% for scientists.

[1] Department of Computer Science, Masdar Institute, Khalifa University of Science and Technology, Abu Dhabi, P.O. Box 54224, UAE. [2] Computer Science, New York University, Abu Dhabi, P.O. Box 129188, UAE. [3] Expedia Inc., 333 108th AVE NE, Bellevue, WA 98004, USA. Correspondence and requests for materials should be addressed to B.K.A. (email: bedoor@deeplearn.net) or to T.R. (email: talal.rahwan@nyu.edu) or to W.L.W. (email: wlwoon@deeplearn.net)

Diversity is highly valued in modern societies[1–6]. Social cohesion, tolerance, and integration are linked to tangible benefits including economic vibrancy[7,8] and innovativeness[5,9–11]. Far from being an abstract ideal, this conviction has guided many governmental and hiring policies and can have broad and long-lasting effects on society[12,13]. However, diversity is a complex issue, as groups can be diverse in terms of various attributes, such as ethnicity, gender, age, and socio-economic background. It is also unclear if all forms of diversity are beneficial. For instance, ethnic density has been associated with positive outcomes in terms of health[14,15], while ethnic polarization has a negative effect on economic development[16]. Furthermore, diversity can be a divisive topic that is clouded by emotion, partisan loyalties, and political correctness, all of which can hinder impartial discussions[17]. The factors above strongly motivate an objective study on the value of diversity, and on whether more diverse groups achieve greater success.

One domain in which this question can be effectively addressed is academia[18,19]. The structure of academic collaboration is observable via co-authorships, which frequently involve scientists from different locations, disciplines and backgrounds[20,21]. Furthermore, academic output has an objective, widely accepted measure—citation count[22,23]. This amenability to analysis has already attracted attempts at identifying the factors which underlie success in academia, an enterprise known as the Science of Science[24]. Although many such factors have been studied, including gender[25], academic age[26], team size[27], inter-disciplinarity[28], ethnicity[29], and affiliation[30,31], the study of these factors is extremely complex and many questions remain unanswered.

Our study seeks to address this shortcoming from a number of hitherto unexplored perspectives. Firstly, we compare homophily in scientific collaborations from the perspectives of age, gender, affiliation, and ethnicity. We find clear signs of homophily in the cases of ethnicity, gender, and affiliation. However, in only one case, ethnicity, was homophily was found to be increasing steadily over time. Secondly, we examine the relationship between various classes of diversity and research impact at the level of scientific fields. Remarkably, we found that ethnic diversity is most strongly associated with scientific impact. Thirdly, we compare the benefits of diversity on groups vs. individuals, and find that the former outweighs the latter. Finally, we study the evolution and effect of diversity over time, team size, and number of collaborators, and verify that the above findings persist across all of these dimensions. The results of these multiple angles of analysis are combined to form a far richer picture of diversity than has been possible in the past.

## Results

**Exploring homophily.** A natural starting point for our study of diversity is to establish the extent to which homophily[32] exists in academia—i.e., whether scientists tend to collaborate more frequently with similar others—which would lead to an overall lack of diversity in scientific collaborations. We use the Microsoft Academic Graph dataset (available at: https://www.microsoft.com/en-us/research/project/microsoft-academic-graph/), and analyze 1,045,401 multi-authored papers (see Supplementary Figure 1 for the distribution of papers by year), written by 1,529,279 scientists, spanning eight main fields and 24 subfields of science. We analyzed diversity in terms of these five attributes: ethnicity (eth), discipline (dsp), gender (gen), affiliation (aff), and academic age (age); see Supplementary Note 1. Here, the abbreviations in parentheses are used in subsequent mathematical expressions to indicate the associated attribute. These attributes reflect many technical and social factors that

influence teamwork and collaboration. Affiliation indicates the geographic location, and may even reflect the way collaborative work is carried out—from the style and culture of collaboration to its mundane details, such as the medium used to collaborate, e.g., face-to-face interactions vs. telecommunication or email. Academic age is not only indicative of the amount of experience that a scientist has, but is also typically associated with actual age. Discipline may reflect a scientist's substantive knowledge and his/her acquired skills through training, as well as the culture in which collaborative work is carried out. Finally, ethnicity and gender may play a role in shaping scientists' social identities, knowledge, and biases. To quantify diversity in terms of any of the aforementioned attributes, we use the Gini Impurity[33], resulting in the following group diversity indices, $d_{\text{eth}}^{\text{G}}$, $d_{\text{age}}^{\text{G}}$, $d_{\text{gen}}^{\text{G}}$, $d_{\text{dsp}}^{\text{G}}$ and $d_{\text{aff}}^{\text{G}}$ (an alternative diversity measure was also considered; see Supplementary Note 2 and Supplementary Figure 2).

To explore homophily, we generate different randomized baseline models whereby a particular attribute—be it ethnicity, gender, affiliation, or academic age—is shuffled. For example, in the case of ethnicity, this process is akin to creating a universe in which ethnicity is disregarded in the selection of co-authors, while retaining other criteria. To preserve the conditional distributions of the ethnicities, the shuffling process is constrained to only occur between authors of papers that have the same subfield, publication year, and number of authors; for full details, see Supplementary Note 3. This way, for every paper $p$ in the real dataset, there exists a matching paper $p'$ in the randomized dataset that may differ from $p$ in terms of ethnic diversity, but is identical to $p$ in terms of gender, affiliation, academic age, citations, publication year, and number of authors per paper. Importantly, while such a baseline model may produce homogeneous groups, the emergence of such groups is purely the result of random chance rather than homophily. As such, by comparing the real dataset with this baseline model, we can determine whether homophily exists, and if so, quantify the degree to which it is spread across academia. Figure 1a compares our real dataset with the randomized baseline model in terms of the cumulative distributions of $d_x^{\text{G}} : x \in \{\text{eth, age, gen, aff}\}$. As can be seen, for $x \in \{\text{eth, gen, aff}\}$, groups with low $d_x^{\text{G}}$ are more common in reality than would be expected by random chance, highlighting the fact that homophily does indeed exist in academia in terms of ethnicity, gender, and affiliation. However, for $x = \text{age}$, the opposite was observed (see Supplementary Figures 3–6 for subfield-specific distributions). These observations persist, regardless of the publication year (Fig. 1b), and the number of authors per paper (Fig. 1c). The temporal trends observed in Fig. 1b are particularly intriguing. For $d_{\text{eth}}^{\text{G}}$, while the population of scientists is becoming more ethnically diverse (see the steady increase in the red line), this trend is not reflected in the actual coauthor groupings, implying that ethnic homophily is steadily increasing. For $d_{\text{age}}^{\text{G}}$, the actual level of diversity is greater than would be expected by random chance; this pattern is regularly observed in academia, e.g., consider the many publications resulting from advisor–advisee collaborations. For $d_{\text{gen}}^{\text{G}}$, although gender homophily continues to exist, it steadily decreases over time, suggesting that women are playing an ever greater role in scientific endeavors. Finally, for $d_{\text{aff}}^{\text{G}}$, there is a marked decrease in affiliation homophily around the 1990s; this is consistent with the jump in multi-university collaborations in the 1990s due to the widespread of the Internet and other technologies that facilitate collaboration across geographically distant scientists[30].

**The link between diversity and scientific impact.** Having explored homophily in academia, we now study the effects of

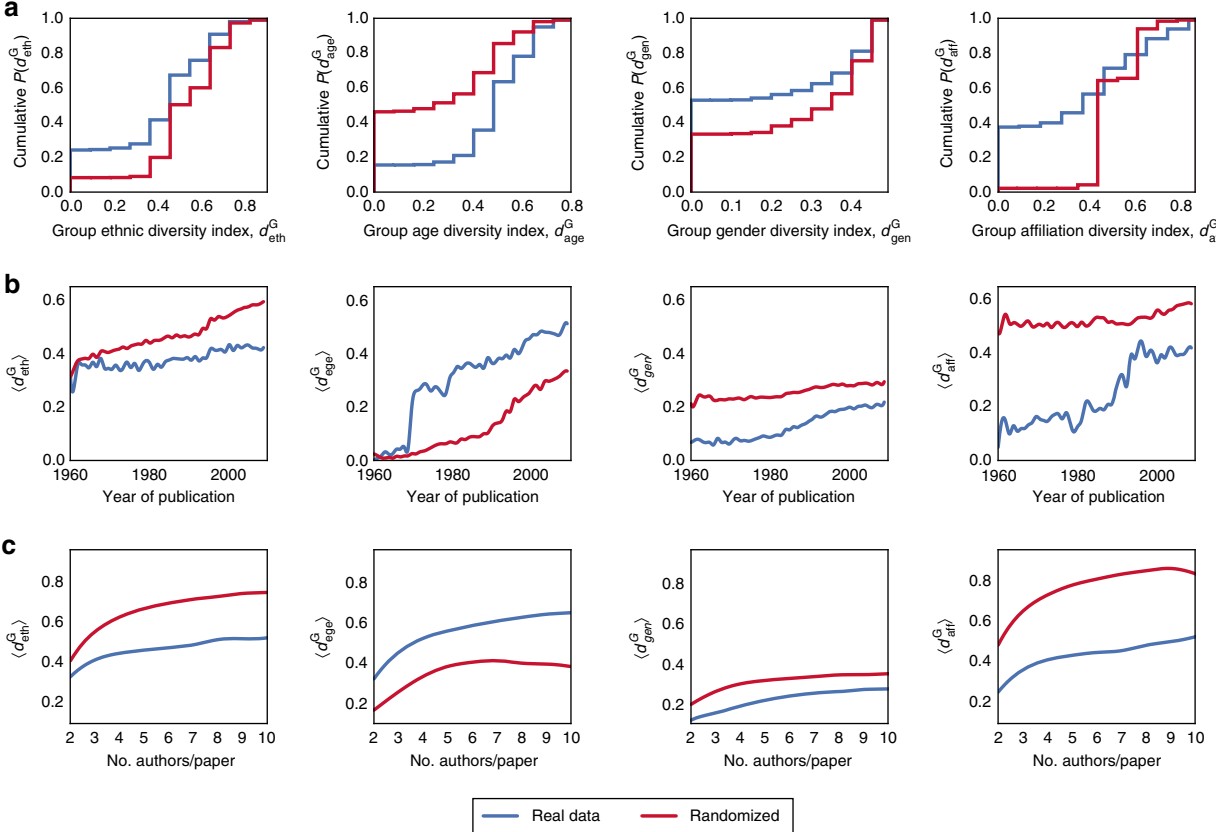

**Fig. 1** Exploring homophily in real vs. randomized data. Each column corresponds to a different class of diversity, and each row presents the results of a specific set of experiments whereby $d_x^G : x \in \{\text{eth, age, gen, aff}\}$ in real data is compared against randomized data. **a** Cumulative distributions of $d_x^G$. **b** Change in mean diversity $\langle d_x^G \rangle$ over time. **c** Mean diversity $\langle d_x^G \rangle$ for papers with different number of authors

homophily (and diversity) on research impact, measured by the number of citations received within 5 years of publication, denoted by $c_5^G$ (see Supplementary Note 4 and Supplementary Figure 7). Using the same dataset and notation described earlier, we study the relationship between a subfield's diversity and its academic impact. Here, we distinguish between two notions of diversity. The first is where the unit of analysis is a paper's set of authors, while the second is where the unit of analysis is an individual scientist's entire set of collaborators. We refer to the former as group diversity, and to the latter as individual diversity; see Fig. 2 for an illustration comparing the two notions.

For each subfield, Fig. 3a depicts the mean group diversity indices, $\langle d_x^G \rangle : x \in \{\text{eth, age, gen, dsp, aff}\}$, against the mean 5-year citation count, $\langle c_5^G \rangle$, taken over papers in that subfield (notation summary and formal definitions are in Supplementary Table 1 and Supplementary Note 2, respectively). Remarkably, we find that a subfield's ethnic diversity is the most strongly correlated with impact ($r = 0.77$); the positive correlation persists even when the subfields are studied in isolation (Supplementary Figures 8 and Supplementary Table 2), regardless of the number of authors per paper (Supplementary Figure 9). These findings are further supported by the regression analysis in Table 1. While these findings do not imply causation, it is still suggestive that one can largely predict scientific impact based solely on average ethnic diversity, especially given that ethnicity is arguably unrelated to technical competence.

Having studied group diversity, we now move our attention to individual diversity. Here, we analyze scientists with at least 10 collaborators each, amounting to a total of 5,103,877 collaborators over 9,472,439 papers (see Supplementary Table 3 for a

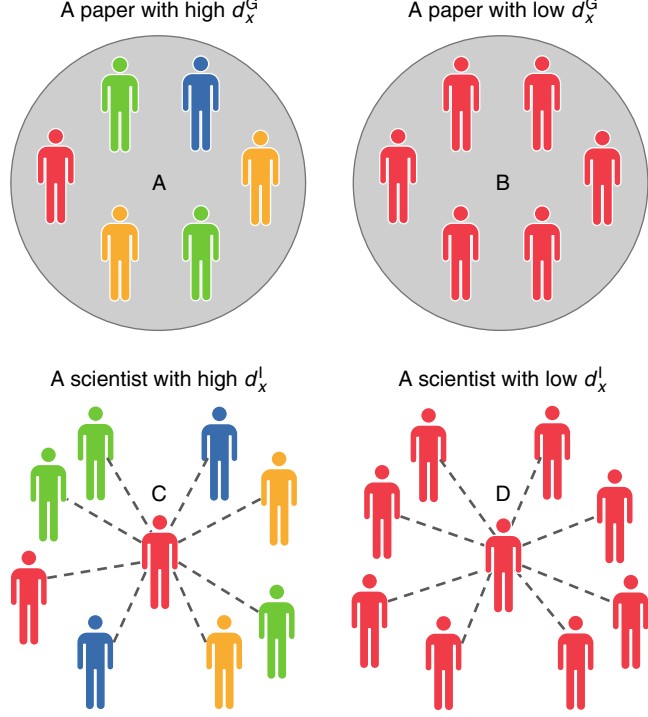

**Fig. 2** Group vs. individual diversity. For any given class of diversity, $x \in \{\text{eth, age, gen, dsp, aff}\}$, differences in color represent differences in terms of $x$. The group diversity index $d_x^G$ of Paper A is higher than that of Paper B. The individual diversity index of Scientist C is higher than that of Scientist D

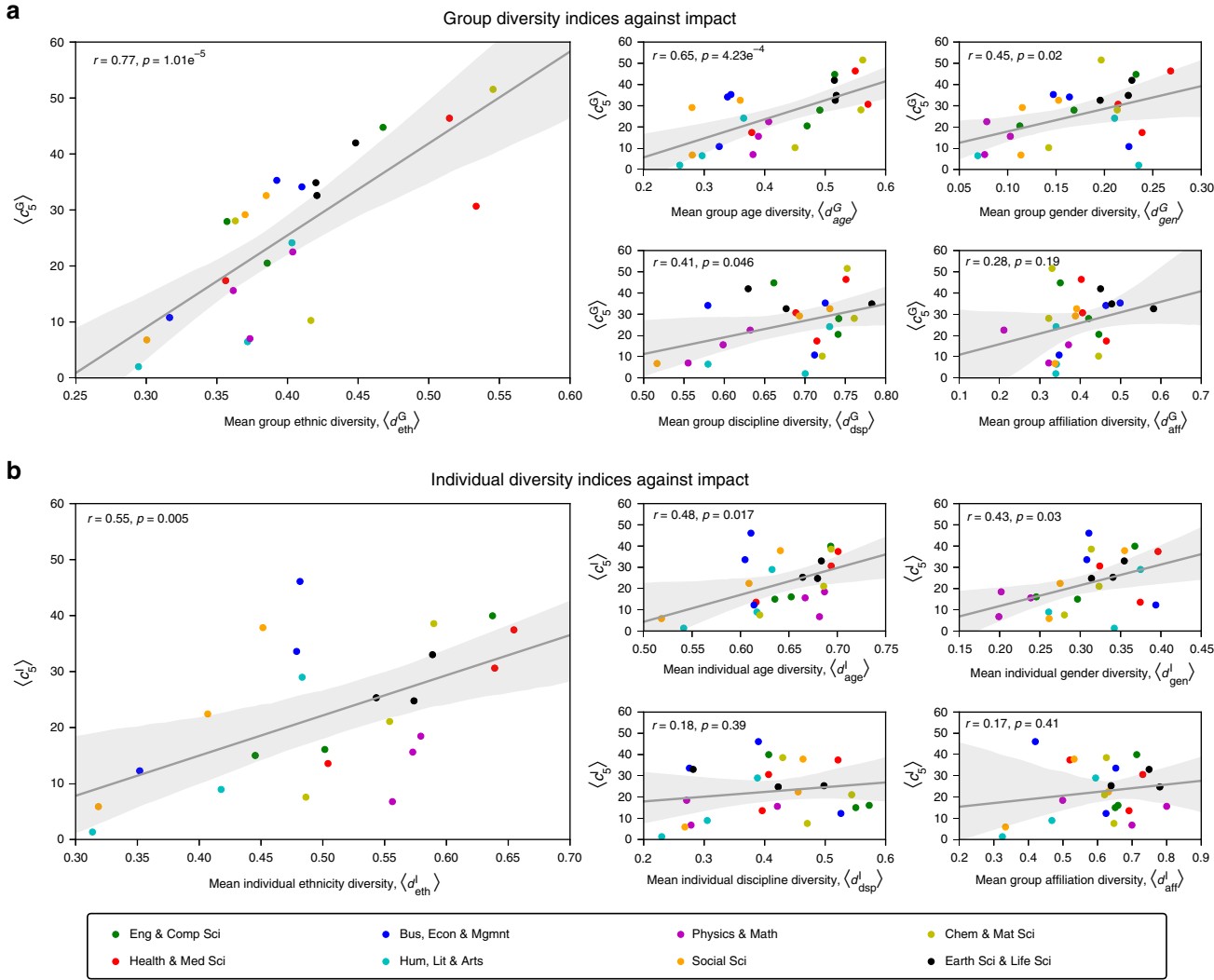

**Fig. 3** Group and individual diversity vs. impact in each subfield. In each subplot, the points correspond to subfields, the color indicates the main field, while the solid line and the shaded area represent the regression line and the 95% confidence interval, respectively. Each regression has also been annotated with the corresponding Pearson's $r$ and $p$ values. **a** For each subfield, the subplots depict the mean group diversity indices, $\langle d_{eth}^G \rangle$, $\langle d_{age}^G \rangle$, $\langle d_{gen}^G \rangle$, $\langle d_{dsp}^G \rangle$ and $\langle d_{aff}^G \rangle$, against the mean 5-year citation count, $\langle c_5^G \rangle$, taken over papers in that subfield. **b** For each subfield, the subplots depict the mean individual diversity indices, $\langle d_{eth}^I \rangle$, $\langle d_{age}^I \rangle$, $\langle d_{gen}^I \rangle$, $\langle d_{dsp}^I \rangle$ and $\langle d_{aff}^I \rangle$, against the mean 5-year citation count, $\langle c_5^I \rangle$, taken over scientists in that subfield

summary of all filters applied on the dataset). For each subfield, Fig. 3b depicts the mean individual diversity indices, $\langle d_x^I \rangle : x \in \{eth, age, gen, dsp, aff\}$, against the mean 5-year citation count, $\langle c_5^I \rangle$, taken over scientists in that subfield. As can be seen, a subfield's ethnic diversity is again the most strongly correlated with impact ($r = 0.55$), even when the subfields are studied in isolation (Supplementary Figure 10 and Supplementary Table 4).

The above results highlight a potential dysfunction. While homophily was observed for ethnicity, affiliation and gender, the only attribute for which it was found to be increasing over time was ethnicity, which seems strange given the apparent pre-eminence of ethnic diversity. Motivated by this observation, we further explore the relationship between ethnic diversity and scientific impact in the randomized universe used earlier in Fig. 1. Recall that, in such a universe, ethnicity is excluded as a criterion for selecting co-authors while the other factors are preserved. Hence, it stands to reason that any differences in impact between the randomized and real datasets can be attributed to ethnic diversity. To examine these differences, we partitioned the papers

into two categories, labeled as diverse $\left( d_{eth}^G > \tilde{d}_{eth}^G \right)$ and non-diverse $\left( d_{eth}^G \leq \tilde{d}_{eth}^G \right)$, where the tilde denotes the median. The scientists were similarly partitioned into diverse $\left( d_{eth}^I > \tilde{d}_{eth}^I \right)$ and non-diverse $\left( d_{eth}^I \leq \tilde{d}_{eth}^I \right)$. We find that the diverse consistently outperforms the non-diverse, regardless of the year of publication (Fig. 4e), the number of authors per paper (Fig. 4g), and the number of collaborators per scientist (Fig. 4i). We replicated these plots using the randomized, instead of the real, dataset (Fig. 4f, h and j). As can be seen, the performance gap between the diverse and non-diverse almost entirely disappears in the randomized dataset, suggesting that the observed impact gains in the real dataset could indeed be attributed to ethnic diversity. Note that, in the real dataset, a large proportion of papers have $d_{eth}^G = 0$ (see Fig. 4a), and a large proportion of scientists have $d_{eth}^I = 0$ (see Fig. 4c). As such, the observed performance gap between the diverse and the non-diverse could be predominantly due to these papers and scientists being less impactful than their counterparts

**Table 1 Regression analyses of diversity classes on academic impact**

| | Citation count, $c_5^G$ | | | | | | | |
|---|---|---|---|---|---|---|---|---|
| | Engineering & computer science | Health & medical sciences | Business, economics & management | Humanities, literature & arts | Physics & mathematics | Social sciences | Chemical & material sciences | Life sciences & earth sciences |
| **(A) Group ethnic diversity** | | | | | | | | |
| $d_{eth}^G$ | 7.40*** | 3.00*** | 5.21*** | 4.77*** | 8.04** | 4.39** | 4.29** | 3.94*** |
| | (2.44) | (0.64) | (1.64) | (1.79) | (3.30) | (1.89) | (1.95) | (1.45) |
| University ranking | −1.22*** | −1.08*** | −0.60** | −0.52** | −0.16 | −0.55* | −0.35 | −1.35*** |
| | (0.39) | (0.08) | (0.24) | (0.26) | (0.46) | (0.29) | (0.29) | (0.23) |
| Author's prior impact | 0.62*** | 1.24*** | 1.52*** | 1.61*** | 0.72*** | 1.51*** | 1.60*** | 1.53*** |
| | (0.01) | (0.01) | (0.01) | (0.01) | (0.02) | (0.01) | (0.01) | (0.01) |
| Year of publication | 0.20 | 0.24*** | 0.07 | 0.48*** | 0.13 | 0.37** | 0.24 | 0.24*** |
| | (0.21) | (0.01) | (0.10) | (0.10) | (0.16) | (0.17) | (0.17) | (0.01) |
| Number of authors | 0.00 | 0.59*** | 0.23 | 0.27 | 1.06 | 0.46** | 0.51*** | 0.69*** |
| | (0.27) | (0.15) | (0.17) | (0.18) | (1.03) | (0.19) | (0.19) | (0.11) |
| Const | 2221.02*** | 598.55*** | 1081.71*** | 1085.84*** | 1289.91*** | 2142.17*** | 1813.42*** | 2750.75*** |
| | (270.36) | (22.94) | (114.27) | (124.13) | (230.16) | (194.14) | (188.89) | (144.35) |
| $R^2$ | 0.11 | 0.24 | 0.33 | 0.35 | 0.19 | 0.34 | 0.35 | 0.39 |
| $N$ | 139705 | 288827 | 38938 | 47141 | 146574 | 158479 | 88300 | 137437 |
| **(B) Group age diversity** | | | | | | | | |
| $d_{age}^G$ | 0.59 | 8.45*** | 15.06*** | 19.82*** | 10.92*** | 23.23*** | 11.41*** | 11.28*** |
| | (3.41) | (0.71) | (1.52) | (2.73) | (3.37) | (3.38) | (2.44) | (1.95) |
| University ranking | −1.41*** | −1.04*** | −0.60** | −0.51** | −0.10 | −0.55* | −0.34 | −1.31*** |
| | (0.39) | (0.08) | (0.24) | (0.26) | (0.46) | (0.29) | (0.30) | (0.23) |
| Author's prior impact | 0.62*** | 1.24*** | 1.52*** | 1.61*** | 0.72*** | 1.51*** | 1.60*** | 1.53*** |
| | (0.01) | (0.01) | (0.01) | (0.01) | (0.02) | (0.01) | (0.01) | (0.01) |
| Year of publication | 0.22 | 0.28*** | 0.38*** | 0.04 | 0.08 | 0.42* | 0.14* | 1.09*** |
| | (0.21) | (0.01) | (0.09) | (0.07) | (0.16) | (0.23) | (0.09) | (0.11) |
| Number of authors | 0.18 | 0.24 | 0.17*** | −0.02 | 0.74 | 0.00 | 0.63 | 0.56*** |
| | (0.28) | (0.15) | (0.06) | (0.76) | (1.04) | (0.21) | (0.49) | (0.12) |
| Const | 2221.02*** | 598.55*** | 1081.71*** | 1085.84*** | 1289.91*** | 2142.17*** | 1813.42*** | 2750.75*** |
| | (270.36) | (22.94) | (114.27) | (124.13) | (230.16) | (194.14) | (188.89) | (144.35) |
| $R^2$ | 0.11 | 0.24 | 0.32 | 0.31 | 0.19 | 0.34 | 0.32 | 0.38 |
| $N$ | 139,705 | 288,827 | 38,938 | 47,141 | 146,574 | 158,479 | 88,300 | 137,437 |
| **(C) Group gender diversity** | | | | | | | | |
| $d_{gen}^G$ | −6.34 | −0.93 | 0.57 | 1.54 | 1.55 | −0.24 | 6.34** | −0.85 |
| | (4.48) | (1.38) | (1.67) | (3.38) | (4.41) | (2.60) | (2.93) | (2.09) |
| University ranking | −0.75 | −0.69*** | 0.06 | −1.72*** | −0.11 | −0.68** | −1.11*** | −0.92*** |
| | (0.56) | (0.12) | (0.19) | (0.41) | (0.59) | (0.29) | (0.35) | (0.29) |
| Author's rior impact | 1.33*** | 1.67*** | 0.92*** | 1.53*** | 0.65*** | 1.47*** | 1.06*** | 1.61*** |
| | (0.02) | (0.02) | (0.01) | (0.04) | (0.03) | (0.01) | (0.05) | (0.01) |
| Year of publication | 0.70** | 0.22*** | 0.34*** | 0.07 | 0.02 | 0.22 | 0.05 | 1.04*** |
| | (0.35) | (0.03) | (0.10) | (0.08) | (0.21) | (0.23) | (0.10) | (0.15) |
| Number of authors | −0.13 | 0.79*** | 0.38*** | 1.44* | 1.75 | 1.12*** | 1.13** | 0.76*** |
| | (0.36) | (0.19) | (0.06) | (0.78) | (1.27) | (0.19) | (0.51) | (0.13) |
| Const | 946.57** | 541.77*** | 2617.85*** | 468.14*** | 1579.15*** | 2669.66*** | 784.17*** | 2787.59*** |
| | (409.64) | (41.14) | (104.67) | (116.18) | (304.95) | (235.49) | (133.25) | (183.41) |
| $R^2$ | 0.16 | 0.29 | 0.32 | 0.31 | 0.17 | 0.39 | 0.26 | 0.41 |
| $N$ | 58,288 | 188,249 | 14,904 | 8911 | 36,949 | 30,420 | 50,887 | 71,630 |
| **(D) Group affiliation diversity** | | | | | | | | |
| $d_{aff}^G$ | −2.85 | 2.93*** | 2.45** | 0.85 | 9.88*** | 5.77*** | 0.43 | 3.89*** |
| | (2.35) | (0.60) | (0.97) | (2.70) | (3.35) | (1.97) | (2.26) | (1.36) |
| University ranking | −1.35*** | −1.16*** | −0.12 | −1.29*** | −0.26 | −0.59** | −0.79*** | −1.42*** |
| | (0.39) | (0.08) | (0.18) | (0.36) | (0.46) | (0.30) | (0.30) | (0.24) |
| Author's prior impact | 0.62*** | 1.23*** | 0.92*** | 1.49*** | 0.72*** | 1.60*** | 1.04*** | 1.53*** |
| | (0.01) | (0.01) | (0.01) | (0.03) | (0.02) | (0.01) | (0.04) | (0.01) |
| Year of publication | 0.14 | 0.25*** | 0.28*** | 0.13* | 0.10 | 0.58** | 0.06 | 1.04*** |
| | (0.21) | (0.01) | (0.09) | (0.07) | (0.16) | (0.23) | (0.09) | (0.11) |
| Number of authors | 0.26 | 0.55*** | 0.35*** | 1.59** | 0.71 | 0.31 | 1.24** | 0.64*** |
| | (0.28) | (0.15) | (0.06) | (0.77) | (1.05) | (0.21) | (0.49) | (0.12) |
| Const | 2240.33*** | 622.76*** | 2370.64*** | 327.82*** | 1336.28*** | 2319.30*** | 793.64*** | 2721.82*** |
| | (275.40) | (23.59) | (91.50) | (97.70) | (230.77) | (231.07) | (117.89) | (144.24) |
| $R^2$ | 0.11 | 0.24 | 0.32 | 0.30 | 0.20 | 0.35 | 0.25 | 0.39 |
| $N$ | 38,236 | 35,925 | 4736 | 2738 | 61,898 | 6431 | 25,656 | 32,279 |
| **(E) Group discipline diversity** | | | | | | | | |
| $d_{dsp}^G$ | 7.39 | 15.08*** | 6.92 | 31.35*** | 24.35*** | 7.00 | 25.05*** | 15.77*** |
| | (9.91) | (1.66) | (5.47) | (6.68) | (7.37) | (13.70) | (7.08) | (3.42) |
| University ranking | −2.46*** | −1.01*** | −0.49 | −1.36*** | −0.96 | 0.28 | −0.85* | −1.75*** |
| | (0.55) | (0.10) | (0.30) | (0.51) | (0.64) | (0.53) | (0.48) | (0.32) |
| Author's prior impact | 0.62*** | 1.35*** | 0.91*** | 1.45*** | 0.69*** | 1.80*** | 0.96*** | 1.55*** |
| | (0.01) | (0.01) | (0.01) | (0.04) | (0.03) | (0.02) | (0.05) | (0.01) |
| Year of publication | 0.15 | 0.28*** | 0.29*** | 0.01 | −0.01 | 0.71*** | 0.19** | 1.13*** |
| | (0.22) | (0.02) | (0.09) | (0.08) | (0.18) | (0.25) | (0.10) | (0.11) |
| Number of authors | 0.02 | 0.05*** | 0.02*** | 0.24* | 0.28 | 0.10*** | 0.17*** | 0.04*** |
| | (0.02) | (0.02) | (0.01) | (0.14) | (0.20) | (0.03) | (0.05) | (0.01) |
| Const | −253.60 | 566.42*** | 598.47*** | 24.34 | 76.50 | −1412.96*** | 387.01** | 2278.47*** |
| | (446.69) | (32.93) | (182.43) | (161.50) | (352.32) | (502.55) | (190.31) | (226.61) |
| $R^2$ | 0.10 | 0.25 | 0.26 | 0.29 | 0.18 | 0.35 | 0.21 | 0.38 |
| $N$ | 104,088 | 141,917 | 20,801 | 12,238 | 100,839 | 24,773 | 65,607 | 98,006 |

The regression tables below present the effect of each of the five group diversity indices, $d_x^G : x \in \{eth, age, gen, aff, dsp\}$, on the paper's impact, $c_5^G$. Along with each class of diversity, the following predictor variables were used: university ranking, author's prior impact, year of publication, and number of authors. Here, university rankings are based on the 2017 Academic Ranking of World Universities, also known as the Shanghai Ranking, whereas an author's prior impact is measured as the annual number of citations that he/she accumulated prior to the year in which the paper was published. The columns correspond to papers from different fields. Of the five classes of diversity studied, ethnic diversity (A) was the only one for which all coefficients in the first row ($d_{eth}^G$) are positive and significant. Standard errors in parentheses
*$p < 0.1$; **$p < 0.05$; and ***$p < 0.01$

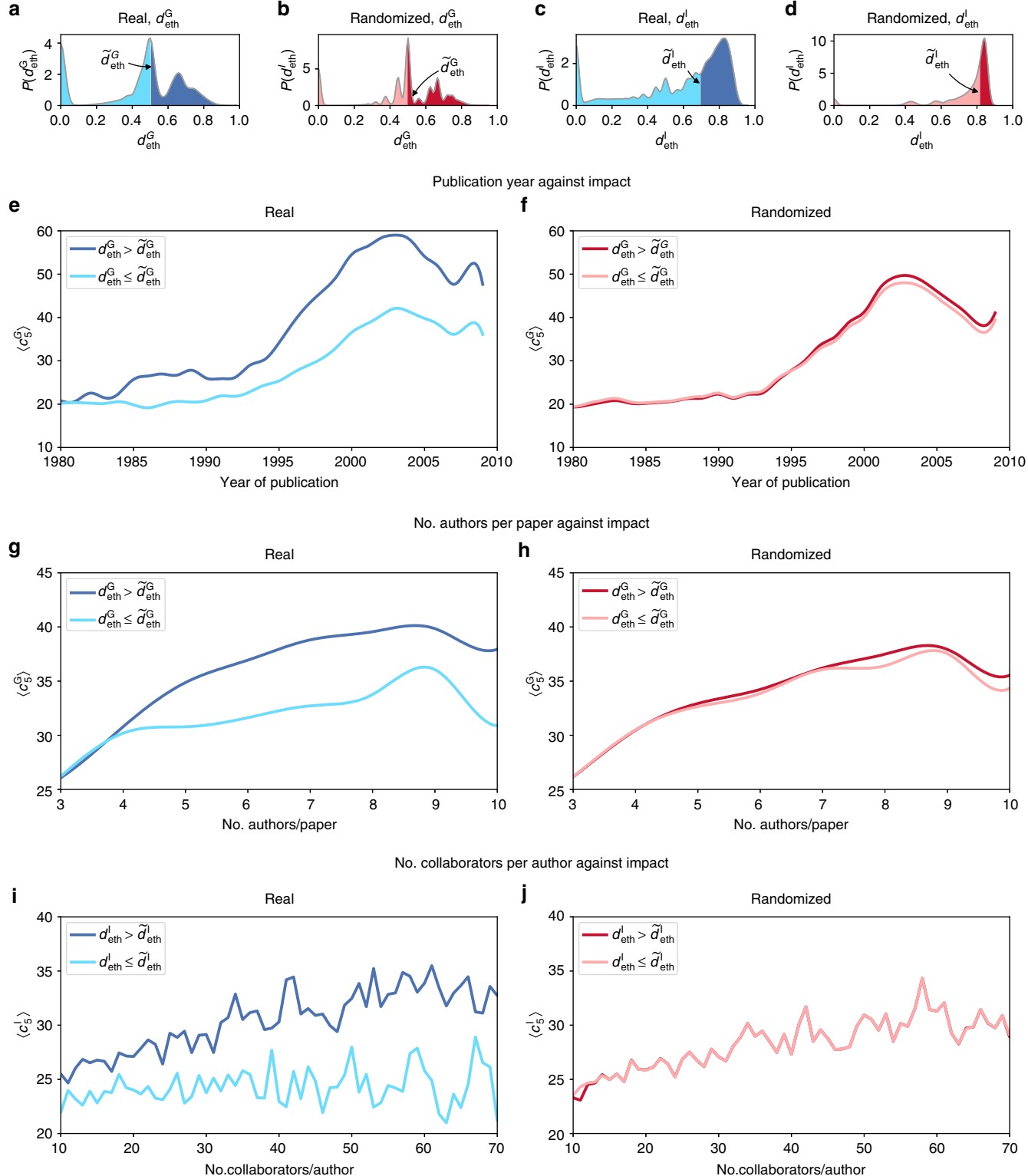

**Fig. 4** The relationship between ethnic diversity and impact. **a** Distribution of $d_{eth}^G$ in real data. Papers were partitioned into two categories: diverse (highlighted in the darker tones, with $d_{eth}^G > \tilde{d}_{eth}^G$) and non-diverse (highlighted in the lighter tones, with $d_{eth}^G \leq \tilde{d}_{eth}^G$), where the tilde denotes the median. **b** The same as (**a**), but for randomized data. **c** and **d** The same as (**a**, **b**), respectively, but with $d_{eth}^I$ instead of $d_{eth}^G$. **e** $\langle c_5^G \rangle$ against publication year in real data. **f** The same as (**e**), but for randomized data. **g** $\langle c_5^G \rangle$ against number of authors per paper in real data. **h** The same as (**g**), but for randomized data. **i** $\langle c_5^I \rangle$ against number of collaborators per scientist in real data. **j** The same as (**i**), but for randomized data

whose $d_{eth}^G > 0$ and $d_{eth}^I > 0$, respectively. To determine whether this is the case, we replicated the analysis of papers but after excluding those with $d_{eth}^G = 0$, and likewise replicated the analysis of scientists but after excluding those with $d_{eth}^I = 0$; see Supplementary Figure 11. As can be seen, even after this exclusion, the diverse mostly outperform the non-diverse, regardless of publication year, number of authors per paper, and number of collaborators per scientist.

**Inferring causality**. To provide further evidence of the link between ethnic diversity and scientific impact, we use coarsened exact matching[34], a technique typically used to infer causality in observational studies[35]. Specifically, it matches the control and treatment populations with respect to the confounding factors identified, thereby eliminating the effect of these factors on the phenomena under investigation. In our case, when studying group ethnic diversity, the treatment set consists of papers for which $d_{eth}^{G} > P_{100-i}(d_{eth}^{G})$, and the control set of papers for which $d_{eth}^{G} \leq P_{i}(d_{eth}^{G})$, where $P_{i}(d_{eth}^{G})$ denotes the $i$th percentile of $d_{eth}^{G}$. This process is repeated using $i = 10, 20, 30, 40, 50$, corresponding to progressively larger gaps in ethnic diversity between the two populations. Thus, if ethnic diversity is indeed associated with increased scientific impact, we would expect to find a significant difference in impact between the two populations, and expect this difference to increase in tandem with the aforementioned gap in diversity. The confounding factors identified were the year of publication, number of authors, field of study, authors' impact prior to publication, and university ranking. The same process was carried out for individual ethnic diversity, for which the confounding factors were academic age, number of collaborators, discipline, and university ranking; see Supplementary Note 5 and Supplementary Figures 12 and 13 for more details, and Supplementary Figure 14 for an illustration of how this process works on a given collection of papers. The results for group and individual ethnic diversities are summarized in Tables 2 and 3, respectively. As can be seen, increasing the diversity gap between the control and treatment populations is

often accompanied by a greater difference in scientific impacts between the two populations. Remarkably, in the case of papers and scientists above the 90th percentile, the difference in scientific impact reaches 10.63% and 47.67%, respectively, compared to their counterparts below the 10th percentile. Clearly, these results do not suggest that diversity is the only causal factor. For example, one may argue that highly ranked universities tend to attract students from around the world and are more ethnically diverse as a result; indeed we verified that this was the case (see Supplementary Note 6 and Supplementary Figures 15 and 16). In such situations, coarsened exact matching is particularly useful precisely because it allows us to establish causality despite such effects.

**Interplay between group and individual ethnic diversity**. Finally, we investigate the interplay between group ethnic diversity, $d_{eth}^{G}$, and individual ethnic diversity, $d_{eth}^{I}$. To this end, for each of the 1,045,401 papers in our dataset, we calculate $d_{eth}^{I}$ averaged over the authors in that paper; we denote this as $\langle d_{eth}^{I} \rangle_{paper}$. This allows us to study the ways in which the two notions of diversity vary in the same paper. Indeed, as illustrated in Fig. 5, a paper can have high $d_{eth}^{G}$ and at the same time have low $\langle d_{eth}^{I} \rangle_{paper}$, and vice versa. With this in mind, we studied the impact, $\langle c_5^{G} \rangle$, of papers falling in different ranges of $d_{eth}^{G}$ and $\langle d_{eth}^{I} \rangle_{paper}$; see the matrix at the bottom-right corner of Fig. 5. Here, if we denote this matrix by $A$, and label the bottom row and leftmost column as 1, we find

### Table 2 Coarsened exact matching of group ethnic diversity

| | $|T|$ | $|C|$ | $|T'|$ | $|C'|$ | $\mathcal{L}_1$ | $\delta$ | $CI_{0.95}$ | $p$ |
|---|---|---|---|---|---|---|---|---|
| $T: d_{eth}^{G} > P_{90}(d_{eth}^{G})$ <br> $C: d_{eth}^{G} \leq P_{10}(d_{eth}^{G})$ | 17,802 | 45,710 | 13,530 | 16,008 | 0.39 | 10.63 | [8.10, 12.38] | 0.003 |
| $T: d_{eth}^{G} > P_{80}(d_{eth}^{G})$ <br> $C: d_{eth}^{G} \leq P_{20}(d_{eth}^{G})$ | 24,827 | 45,710 | 18,965 | 16,165 | 0.38 | 10.22 | [8.12, 12.02] | 0.0009 |
| $T: d_{eth}^{G} > P_{70}(d_{eth}^{G})$ <br> $C: d_{eth}^{G} \leq P_{30}(d_{eth}^{G})$ | 56,662 | 58,889 | 51,782 | 39,216 | 0.27 | 4.93 | [3.74, 5.97] | 0.008 |
| $T: d_{eth}^{G} > P_{60}(d_{eth}^{G})$ <br> $C: d_{eth}^{G} \leq P_{40}(d_{eth}^{G})$ | 63,129 | 78,340 | 57,279 | 58,199 | 0.29 | 5.14 | [4.12, 6.17] | 0.003 |
| $T: d_{eth}^{G} > P_{50}(d_{eth}^{G})$ <br> $C: d_{eth}^{G} \leq P_{50}(d_{eth}^{G})$ | 63,129 | 127,629 | 58,292 | 70,627 | 0.27 | 3.37 | [2.45, 4.25] | 0.018 |

$T$ and $C$ are the treatment and control populations respectively; $T'$ and $C'$ are the populations of matched treatment and matched control papers, respectively; $\mathcal{L}_1$ is the multivariate imbalance statistic[34]; $\delta$ is the relative impact gain of $T'$ over $C'$, i.e., $\delta = 100 \times \left( \langle c_5^{G} \rangle_{T'} - \langle c_5^{G} \rangle_{C'} \right) / \langle c_5^{G} \rangle_{C'}$. A $t$-test shows that $\delta$ is statistically significant; see the resulting $p$-values. Since the academic impact $\langle c_5^{G} \rangle$ is sensitive to extremal values, we bootstrap a 95% confidence interval ($CI_{0.95}$). Here, university ranking corresponds to the average rank of all universities in the paper, as opposed to the highest ranked university in the paper, which is the case in Supplementary Table 5. For more details, see Supplementary Note 6.

### Table 3 Coarsened exact matching of individual ethnic diversity

| | $|T|$ | $|C|$ | $|T'|$ | $|C'|$ | $\mathcal{L}_1$ | $\delta$ | $CI_{0.95}$ | $p$ |
|---|---|---|---|---|---|---|---|---|
| $T: d_{eth}^{I} > P_{90}(d_{eth}^{I})$ <br> $C: d_{eth}^{I} \leq P_{10}(d_{eth}^{I})$ | 113,883 | 68,563 | 16,512 | 20,599 | 0.47 | 47.67 | [44.49, 49.92] | 2.04e−39 |
| $T: d_{eth}^{I} > P_{80}(d_{eth}^{I})$ <br> $C: d_{eth}^{I} \leq P_{20}(d_{eth}^{I})$ | 139,015 | 136,837 | 65,412 | 50,240 | 0.35 | 43.54 | [42.61, 45.05] | 1.50e−156 |
| $T: d_{eth}^{I} > P_{70}(d_{eth}^{I})$ <br> $C: d_{eth}^{I} \leq P_{30}(d_{eth}^{I})$ | 223,747 | 205,686 | 128,001 | 117,560 | 0.32 | 28.75 | [28.10, 29.46] | 1.65e−211 |
| $T: d_{eth}^{I} > P_{60}(d_{eth}^{I})$ <br> $C: d_{eth}^{I} \leq P_{40}(d_{eth}^{I})$ | 280,514 | 274,209 | 184,749 | 143,683 | 0.29 | 23.86 | [22.86, 23.98] | 5.96e−218 |
| $T: d_{eth}^{I} > P_{50}(d_{eth}^{I})$ <br> $C: d_{eth}^{I} \leq P_{50}(d_{eth}^{I})$ | 356,564 | 329,066 | 242,123 | 240,237 | 0.28 | 15.77 | [15.21, 15.95] | 3.23e−158 |

The notation is as per Table 2.

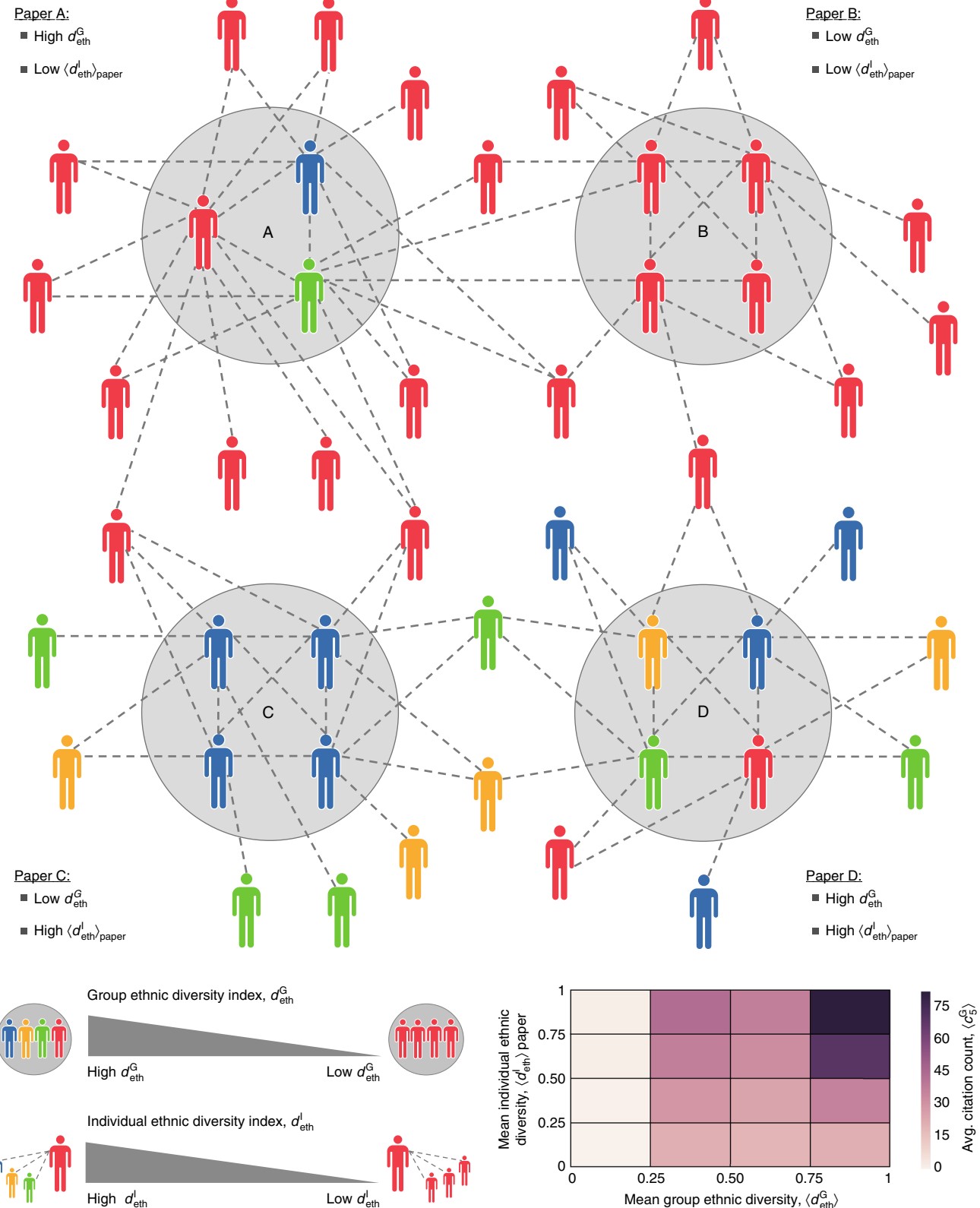

**Fig. 5** The interplay between group and individual ethnic diversity. The top part of the figure illustrates an example of 4 papers. The authors of paper A have different ethnicities, but each has ethnically homogeneous collaborators. Then, one could argue that paper A has high $d_{eth}^G$ but low $\langle d_{eth}^I \rangle_{paper}$. Similarly, paper B has low $d_{eth}^G$ and low $\langle d_{eth}^I \rangle_{paper}$, paper C has low $d_{eth}^G$ and high $\langle d_{eth}^I \rangle_{paper}$, and paper D has high $d_{eth}^G$ and high $\langle d_{eth}^I \rangle_{paper}$. The matrix at the bottom-right corner represents the mean citation counts, $\langle c_5^G \rangle$, of papers falling in different ranges of $d_{eth}^G$ and $\langle d_{eth}^I \rangle_{paper}$

that $\sum_{i=1}^{4} A_{i,1} < \sum_{i=1}^{4} A_{1,i}$ and $\sum_{i=1}^{4} A_{i,4} > \sum_{i=1}^{4} A_{4,i}$. Hence, while it appears that both group and individual diversities can be valuable, the former seems to have a greater effect on scientific impact. In other words, having co-authors who are inclined to collaborate across ethnic lines (i.e., co-authors whose individual ethnic diversity is high) appears to be not as important as the mere presence of co-authors of different ethnicities (i.e., co-authors whose group ethnic diversity is high).

## Discussion

To summarize, this study is the first to cover five different classes of diversity, which allowed us to illuminate many interesting connections between diversity and scientific collaboration. It was also important to establish the occurrence of homophily, and this was achieved via a set of randomized baseline models. These were used to compare observed collaborations with simulated data where the attribute of interest was randomized while controlling for the relevant confounding variables. These comparisons revealed clear and consistent patterns of homophily in the cases of ethnicity, gender, and affiliation, and also revealed that ethnicity was the only attribute for which homophily is increasing over time. In the case of academic age, inverse homophily was found, i.e., scientists seem to prefer collaborating with individuals from different age groups, a possible reflection of the widely held practice of research students being mentored by, and collaborating with, more senior academics.

Armed with these results, we shifted our focus to the effect of homophily (and diversity) on scientific impact. This analysis was conducted using a number of different analytical tools, including regression analysis, randomized baseline models, and coarsened exact matching. Broadly, we found that diversity was positively correlated with impact, though the statistical significance of the observed effect varied significantly depending on the class of diversity and field of study. Overall, discipline and affiliation diversity were the least correlated with impact, a surprising finding given the apparent importance of these attributes. Conversely, ethnic diversity had the strongest correlation, which is especially surprising since ethnicity is not as related to technical competence as the other classes mentioned.

These findings have significant implications. For one, recruiters should always strive to encourage and promote ethnic diversity, be it by recruiting candidates who complement the ethnic composition of existing members, or by recruiting candidates with proven track records in collaborating with people of diverse ethnic backgrounds. Another implication is that, while collaborators with different skill sets are often required to perform complex tasks, multidisciplinarity should not be an end in of itself; bringing together individuals of different ethnicities—with the attendant differences in culture and social perspectives—could ultimately produce a large payoff in terms of performance and impact. To put it differently, intangible factors, such as team cohesion and a sense of esprit de corps should be considered in addition to technical alignment.

The underlying message is an inclusive and uplifting one. In an era of increasing polarization and identity politics, our findings may positively contribute to the societal conversation and reinforce the conviction that good things happen when people of different backgrounds, cultures, and ethnicities come together to work towards shared goals and the common good.

## Data availability

The details of all data and methods used are given in Supplementary Note 1.

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

## Acknowledgements
We thank Steven Skiena and his team for providing access to their Name Ethnicity Classifier tool[36,37]. We also thank Kinga Makovi for suggesting the use of coarsened exact matching for causal inference.

## Author contributions
B.K.A., T.R., and W.L.W. conceived and designed the experiments. B.K.A. and W.L.W. performed the coding of the experiments. B.K.A., T.R., and W.L.W. wrote the manuscript. B.K.A. and T.R. produced the figures and tables.

## Additional information

**Competing interests:** The authors declare no competing interests.

