## [Peer Review File · Nature Communications]

Reviewers' comments:

Reviewer #1 (Remarks to the Author):

This manuscript aims at answerin an interesting question in academic publishing: does "diversity" increase scientific impact?

Diversity is defined in different ways (age, gender, discipline, ethnicity, etc) and impact is measured by citation measures.

After an extensive analysis, diversity, overall, seems to increase the scientific impact.

Among the different types of diversity it seems also that the "ethnic" diversity has the largest impact.

The paper is well referenced and well structured. The analysis has been carried out in a proper way and the results presented clearly.

Nevertheless, I think a couple of points are worth to be investigated.

In the control variables there is no information about the University. Could it be that diversity is correlated with top Universities that also posses larger diversity because they attract very good scholars from every corner of the world? I.e. do we observe endogeneity ?

Although the control variable "Affiliation" is included in the analysis, I think this is mildly different from my question.

A quick answer would be to check for the top ranked university in terms of diversity.

It seems like the correlation coefficient ranges from 0.5 to 0.7 (more or less). Is there some nonlinear effect going on ? I.e. some quadratic relationship between diversity and scientific impact? (do too much diversity compromise quality/impact?). Diversity is also a function of the number of coauthors and I didn't see if, in the manuscript, a normalization per number of authors has been considered or not.

A further analysis should a be a simple Chi-Squared tests, to match the distributions of treated and control groups. The authors use the L_1 imbalance measure (whose values the authors do not comment: are they good or bad?) which is fine (and should be kept in the tables) in the multidimensional case but, I guess, for their analysis a Chi-Squared test can be used.

Overall, this paper is an interesting one, well structured and with a clear evidence of its thesis: ethnic diversity increases impact factor.

Reviewer #2 (Remarks to the Author):

Summary:

This paper quantifies the diversity within scientific collaborations from two perspectives: i) collaborations on an individual paper and ii) collaborations throughout a scientific career. In both cases, the authors interpret their findings to suggest: 1) scientific collaborations are ethnically homophilic and 2) the greater the ethnic diversity, the greater the scientific impact.

Overall the paper offers a novel analysis and adapts several recent advances in the science of science and computational social science to explore a topic of potentially great interest to a broad interdisciplinary audience. The core framework of the paper is technically sound, and lends sufficient evidence to support a potentially controversial result. However, I found the main storyline to be a bit convoluted and there are several technical concerns, as listed below. Therefore, I recommend a major revision before endorsing the article's publication in Nature Communications.

General notes:

It is awkward that the story bounces from diversity vs impact, to homophily, and then back to diversity vs impact. I think it would make more sense to focus on measuring homophily, and then demonstrate its effect on impact. To this end, divide the work into two sections:

1) explore the propensity for homophily in scientific collaboration across multiple diversity indices.

There are many important questions here that should be addressed:

- + Are some disciplines significantly more homophilic than others?
- + How have the diversity indices changed over time (not just ethnicity).
- + Are the differences in diversity across discipline/time a function of population composition (i.e. there is greater gender diversity because the workforce is more balanced) or is it really a change in the propensity for inter-class collaboration?

The random model used in Fig 2 to suggest homophilic collaboration is not sufficiently well documented (page 3, par 2). Were ethnicities shuffled in the entire dataset? This seems like a drastic form of randomness- we know that the population composition is changing in time / discipline for all diversity types. Create another random model in which publication year and discipline are held constant while the other labels are shuffled. Does the case for homophily still exist? Furthermore, when randomizing the ethnicities of authors, is each authorship for each collaborator treated independently? Due to the tendency for repeated collaboration, I would assume this to introduce much more randomness than is found in the data. Here, control for collaborator diversity by randomizing the labels for the individual and distribute this to all papers published by the individual.

2) Go after the question of relating diversity to impact across the multiple diversity types (not just ethnicity).

The strength of this paper rests on the Coarsened Exact Matching experiment. While initially convincing, I would like to see some additional support in the supplemental. Are the covariates actually balanced in the matched groups? Show the mean/distribution for covariates in the matched samples. Are the results of the CEM study robust to the binning decisions? We also know that aggregate citation counts are sensitive to the extremal values, can you bootstrap a confidence interval for comparing random groups of papers with the same sample sizes as the treatment & control groups.

Specific notes:

~ Fig 4 would be more helpful as the first figure of the paper since it illustrates many important concepts used throughout the work (but you should separate out the heat map to a later figure). Furthermore, the colors can denote any of the diversity indices and is not limited to ethnic.

~ The multiple colors are not helpful in Fig 1 without a legend. The result presented in Fig 2 F/G is identical and much easier to read. The caption for Fig 2 F/G does not detail the figure (I'm assuming each point is a subfield).

~ It is more typical to report R^2 values rather than R.

~ It is hard to follow which papers were actually used in the final dataset. Page 3 lists four exclusion criteria, page 10 adds two additional criteria, and page 14 lists even more filters applied to the data. Furthermore, there is a big difference between a scientist who published one paper with 10 co-authors, and one who published 10 papers each with 1 co-author. Were there baselines for minimum number of papers an author wrote when studying a career? The supplemental should better characterize the final dataset analyzed in the work.

~ How were careers re-constructed? How certain are you about the accuracy of this process?

~ The description of Academic Age is confusing. Was it calculated at the year of publication or for all scientists in 2009? The text reads as the latter, which would not capture the true nature of academic age in the given context. I think this is just a writing issue, but please clarify.

~ Fig 3 would be easier to follow if 2 different tones were used to differentiate real vs randomized, and then 2 shades denoted diverse vs non-diverse.

~ While Table 1 and 2 provide sufficient evidence to support the conclusions of the CEM, they do not illustrate the intuition behind the method to an un-experienced reader. I have a feeling this paper might attract many people who are not familiar with CEM. Please provide an illustration of the treatment v control matching process, and highlight the trend that a reader is supposed to identify.

Reviewer #3 (Remarks to the Author):

Review for NCOMMS-18-16552: Ethnic Diversity Increases Scientific Impact

This paper analyzes millions of paper outputs, spanning 24 fields and millions of scientists to examine how different types of diversity (ethnicity, disciplinary, gender, affiliation, and age) correlate with scientific impact. Distinctively, the authors generate novel coding of scientist ethnicities, ages, and gender, and construct (new) diversity measures across both the group and individual level. Overall, this paper's focus on the compositional determinants of scientific performance across a broad swath of academia is welcome and of interest to the Nature Communications readership. However, I found significant concerns with this paper. I outline these concerns below, and hope that you find them helpful.

1. Most importantly, as the authors note, patterns that they present are correlations, not causal. As a result, how meaningful and useful are these univariate correlations? I believe your results of a positive link between ethnic diversity and citation counts. Nonetheless, there are many possible explanations for this link. Off the top of my head, more productive scientists run more international labs, as they reputation allows them to draw across populations. Their prominence could also allow these "stars" to coauthor more broadly. In other words, the correlation could go both ways: scientific impact could result in more ethnically diverse groups. At this stage, it's hard to unpack what is actually going on.

2. You make a number of laudable attempts to drive in the direction of causality. However, I remain unconvinced, while noting that a causal story is incredibly difficult for the type of question you are asking. The randomized comparison set is a bit of a straw-man if you think about it. If I were a skeptic, I would argue that you cherry-picked a result and the randomized sample is merely regression to the mean. For the CEM, the identifying assumption remains a comprehensive set of covariates, and this assumption is untestable.

3. Ultimately, I think it would benefit this paper to stick to its strengths, which is the data itself. There is no need (or expectation) that the patterns you present are causal. Leave that to the small-group experimental psychologists. Merely present this paper as an interesting fact,

presented across a comprehensive window into academia. By claiming less, I think the paper would be more powerful.

4. What is the actual number of publications used? Is it 9 million? 1 million? It's hard to decipher your empirics here.

5. I don't see that much a difference between individual and group diversity (e.g., Figure 4). If we don't see that many differences, is the distinction itself important?

6. I might play up changes in time more. Understanding the changing role of diversity would be particularly interesting.

7. As you might infer, I am not a CS scholar. With all due respect, I might go ahead and run some multiple regressions, which would allow for the necessary controls to be incorporated and for more robust correlations. This also might make the results more easily translated to the social science sphere.

8. Hats off for collecting a great dataset. I do think this is a promising project.

Dear Reviewers,

Thank you for your time and effort reviewing our work and for your comments, which helped us to greatly improve and significantly expand our manuscript. Note that these include several additional analyses, which led to a number of interesting new insights (please see the new version of the manuscript for more details).

Below, we specify our responses (highlighted in blue) to every comment.

Reviewer #1:

This manuscript aims at answerin an interesting question in academic publishing: does "diversity" increase scientific impact?

Diversity is defined in different ways (age, gender, discipline, ethnicity, etc) and impact is measured by citation measures.

After an extensive analysis, diversity, overall, seems to increase the scientific impact.

Among the different types of diversity it seems also that the "ethnic" diversity has the largest impact.

The paper is well referenced and well structured. The analysis has been carried out in a proper way and the results presented clearly.

Nevertheless, I think a couple of points are worth to be investigated.

In the control variables there is no information about the University. Could it be that diversity is correlated with top Universities that also posses larger diversity because they attract very good scholars from every corner of the world? I.e. do we observe endogeneity ?

Although the control variable "Affiliation" is included in the analysis, I think this is mildly different from my question.

A quick answer would be to check for the top ranked university in terms of diversity.

Thank you for pointing out this important issue. This was partially addressed this in the previous version in the Coarsened Exact Matching (CEM) analysis, by adding a confounding factor called "*affiliation ranking*", which represents whether or not one of the affiliations in the paper falls within the top 500 universities according to the *Academic Ranking of World Universities 2017* (also known as the "*Shanghai Ranking*"). Nevertheless, we agree with the importance of the point you raised. To address this point more thoroughly, we revised the manuscript as follows:

- We followed your recommendation to check the top ranked universities in terms of diversity. To this end, we produced four new plots:
 - a. Group ethnic diversity against rankings: 1, 2, ..., 99, 100;
 - b. Group ethnic diversity against ranking bins: 1-100; 101-200; 201-300; 301-400; 401-500.
 - c. Individual ethnic diversity against rankings: 1, 2, ..., 99, 100;
 - d. Individual ethnic diversity against ranking bins: 1-100; 101-200; 201-300; 301-400; 401-500.

When analyzing papers, we controlled for the number of authors, and when analyzing individuals, we controlled for the number of collaborators. The results can be found in the newly-added Section S3, and for convenience we also included the relevant figures below. Indeed, as you anticipated, higher-ranked universities tend to be significantly more ethnically diverse ($p < 0.001$). Nevertheless, even when controlling for university ranking, the relationship between ethnic diversity and scientific impact persists, as will be shown in the next point in

our response.

- We modified the "affiliation ranking" confounding factor in CEM as follows: Firstly, to avoid confusing it with "affiliation diversity", we now call the confounding factor "university ranking". Secondly, instead of having just the following two bins:

1. Ranked in top 500;
2. Not ranked in top 500.

we now have six bins:

1. Ranked in top 100;
2. Ranked in 101-200;
3. Ranked in 201-300;
4. Ranked in 301-400;
5. Ranked in 401-500;

6. Not ranked in top 500.

Furthermore, instead of just focusing on the highest ranked affiliation in the paper, we now consider two alternatives where “university ranking” refers to either:

- (A) the *average ranking* of all affiliations in the paper.
- or
- (B) the *highest ranked* affiliation in the paper;

In contrast, when analyzing individuals, “university ranking” refers to:

- (C) the *ranking* of the affiliation of that individual

The CEM analysis for (A) is in Table 1; for (B) is in Table S3; and for (C) is in Table 2. The new results are very similar to the analysis in the first version of the manuscript, demonstrating the robustness of our findings.

It seems like the correlation coefficient ranges from 0.5 to 0.7 (more or less). Is there some nonlinear effect going on? I.e. some quadratic relationship between diversity and scientific impact? (do too much diversity compromise quality/impact?).

To investigate this point, a quadratic function was fitted to the data instead of the linear regression in Figure 1; the result is depicted below:

As can be seen, the resulting line is monotonic. Nevertheless, while it closely resembles a linear fit, there is a slight but perceptible concavity, suggesting that the impact gain associated with ethnic diversity does not increase indefinitely. These plots have now been added to Supplementary Materials; see Figure S10.

Diversity is also a function of the number of coauthors and I didn't see if, in the manuscript, a normalization per number of authors has been considered or not.

To address this important issue, we generated new plots where we control for the number of authors; the results are shown below. As can be seen, the correlation between the mean ethnic diversity and mean impact per paper in a scientific subfield persists even when controlling for the number of authors. The plots have now been added to Supplementary Materials; see Figure S11.

A further analysis should be a simple Chi-Squared tests, to the distributions of treated and control groups. The authors use the L_1 imbalance measure (whose values the authors do not comment: are they good or bad?) which is fine (and should be kept in the tables) in the multidimensional case but, I guess, for their analysis a Chi-Squared test can be used.

For each confounding factor considered in the CEM, we now compare the distribution of treated and control groups. In particular, we produced two sets of plots: the first concerns the effect of **group** ethnic diversity, and the second concerns that of **individual** ethnic diversity; see the two corresponding figures below. Admittedly, the two populations are not identical, hence the use of the CEM, which is designed to compensate for such differences. These plots have now been added to the Supplementary Materials; see Figures S12 and S13. Note that each subfigure has been annotated with its corresponding Chi-Squared test results, all of which were significant ($p < 0.0001$).

Treatment : $d_{eth}^G > P_{90}(d_{eth}^G)$
 Control : $d_{eth}^G \leq P_{10}(d_{eth}^G)$

Treatment : $d_{eth}^G > P_{50}(d_{eth}^G)$
 Control : $d_{eth}^G \leq P_{50}(d_{eth}^G)$

Treatment : $d_{eth}^l > P_{90}(d_{eth}^l)$
 Control : $d_{eth}^l \leq P_{10}(d_{eth}^l)$

$\chi^2 = 2758.43, p = 0.00$

Treatment : $d_{eth}^l > P_{50}(d_{eth}^l)$
 Control : $d_{eth}^l \leq P_{50}(d_{eth}^l)$

$\chi^2 = 6770.21, p = 0.00$

$\chi^2 = 7825.11, p = 0.00$

$\chi^2 = 30494.70, p = 0.00$

$\chi^2 = 823.89, p = 0.00$

$\chi^2 = 365.77, p = 0.00$

$\chi^2 = 3272.02, p = 0.00$

$\chi^2 = 12109.04, p = 0.00$

Overall, this paper is an interesting one, well structured and with a clear evidence of its thesis: ethnic diversity increases impact factor.

Thank you for your encouragement.

Reviewer #2:

Summary:

This paper quantifies the diversity within scientific collaborations from two perspectives: i) collaborations on an individual paper and ii) collaborations throughout a scientific career. In both cases, the authors interpret their findings to suggest: 1) scientific collaborations are ethnically homophilic and 2) the greater the ethnic diversity, the greater the scientific impact.

Overall the paper offers a novel analysis and adapts several recent advances in the science of science and computational social science to explore a topic of potentially great interest to a broad interdisciplinary audience. The core framework of the paper is technically sound, and lends sufficient evidence to support a potentially controversial result. However, I found the main storyline to be a bit convoluted and there are several technical concerns, as listed below. Therefore, I recommend a major revision before endorsing the article's publication in Nature Communications.

General notes:

It is awkward that the story bounces from diversity vs impact, to homophily, and then back to diversity vs impact. I think it would make more sense to focus on measuring homophily, and then demonstrate its effect on impact. To this end, divide the work into two sections:

Thank you for pointing this out - the manuscript has been extensively restructured in accordance with this recommendation. The story now *starts* with the detection of homophily. Only after the presence of homophily has been established, do we then explore the effects of homophily (and diversity) on research impact as quantified by citation counts.

1) explore the propensity for homophily in scientific collaboration across multiple diversity indices. There are many important questions here that should be addressed:

- + Are some disciplines significantly more homophilic than others?
- + How have the diversity indices changed over time (not just ethnicity).
- + Are the differences in diversity across discipline/time a function of population composition (i.e. there is greater gender diversity because the workforce is more balanced) or is it really a change in the propensity for inter-class collaboration?

To study the prevalence of the different types of homophily (apart from ethnic), additional randomized baseline models were constructed corresponding to each of the four diversity types (note that discipline diversity is excluded from this analysis as the shuffling procedure already groups the authors by field). These models were then used to generate randomized author groupings from which baseline diversity indices were calculated and used for comparison with the actual observed data. These results are now presented in Figure 1 in the updated manuscript. Next, for each diversity type, separate models corresponding to each field are created and, in the same way, used to produce discipline-specific figures for each diversity type. As this produced a very large number of figures, we present these results in the Supplementary Materials (Figures S4 to S7).

From these figures, it can be seen that, while there are slight variations between the different diversity types and fields, the overall picture is that the new results support our original intuition that homophily is indeed a widely observed phenomenon, where the randomized groups were broadly more diverse than the actual author groupings.

The random model used in Fig 2 to suggest homophilic collaboration is not sufficiently well documented (page 3, par 2). Were ethnicities shuffled in the entire dataset? This seems like a drastic form of randomness- we know that the population composition is changing in time / discipline for all diversity types. Create another random model in which publication year and discipline are held constant while the other labels are shuffled. Does the case for homophily still exist? Furthermore, when randomizing the ethnicities of authors, is each authorship for each collaborator treated independently? Due to the tendency for repeated collaboration, I would assume this to introduce much more randomness than is found in the data. Here, control for collaborator diversity by randomizing the labels for the individual and distribute this to all papers published by the individual.

We agree that the shuffling process was not sufficiently documented in the original manuscript. To rectify this, we have now added Section S4, which explains precisely how this process is carried out. Indeed, a poorly-designed shuffling process could suffer from some (or even all) of the potential limitations you have raised. In our case, however, we believe the shuffling process is already designed to address these limitations (note that this is the same shuffling process that was in the original manuscript; we only added Section S4 to describe how this process is carried out, following your recommendation).

2) Go after the question of relating diversity to impact across the multiple diversity types (not just ethnicity).

The strength of this paper rests on the Coarsened Exact Matching experiment. While initially convincing, I would like to see some additional support in the supplemental. Are the covariates actually balanced in the matched groups? Show the mean/distribution for covariates in the matched samples. Are the results of the CEM study robust to the binning decisions? We also know that aggregate citation counts are sensitive to the extremal values, can you bootstrap a confidence interval for comparing random groups of papers with the same sample sizes as the treatment & control groups.

For each confounding factor considered in the CEM, we now compare the distribution of treated and control groups. In particular, we produced two sets of plots: the first concerns the effect of **group** ethnic diversity, and the second concerns that of **individual** ethnic diversity; see the two corresponding figures below. Admittedly, the two populations are not identical, hence the use of the CEM, which is designed to compensate for such differences. These plots have now been added to the Supplementary Materials; see Figures S15 and S16. As per the recommendation of Reviewer 1, each subfigure has been annotated with its corresponding Chi-Squared test results, all of which were significant ($p < 0.0001$). We experimented with other binning decisions and the results were found to be robust. This statement has now been added to Section S5. We also followed your recommendation to bootstrap a (95%) confidence interval, the results of which have now been added to Tables 1, 2 and S3.

Specific notes:

~ Fig 4 would be more helpful as the first figure of the paper since it illustrates many important concepts used throughout the work (but you should separate out the heat map to a later figure). Furthermore, the colors can denote any of the diversity indices and is not limited to ethnic.

In the newly-modified manuscript, we start by studying homophily, followed by studying the relationship between diversity and impact, where group diversity and individual diversity are studied *in isolation*. Finally, we investigate the *interplay* between group and individual diversity (e.g., when a paper has high group diversity and, at the same time, has low individual diversity). Figure 4 illustrates this interplay, and it would seem too early to discuss this at the beginning of the article, especially given your recommendation to start with homophily instead. Therefore, in the new flow of the article, we believe it is more appropriate to leave Figure 4 till the end. We hope the reviewer agrees.

~ The multiple colors are not helpful in Fig 1 without a legend. The result presented in Fig 2 F/G is identical and much easier to read. The caption for Fig 2 F/G does not detail the figure (I'm assuming each point is a subfield).

Yes, in Figure 1 (which is now called Figure 2) each point represent a subfield. To make this clearer, we made the legend more visible by increasing its size, and edited the first sentence in the caption to highlight this point. Figure 2 F/G is no longer an issue, since it has now been omitted to improve the flow of the paper, following the various comments and restructuring suggestions by the reviewers.

~ It is more typical to report R^2 values rather than R .

We agree that reporting R^2 instead of r , as we do in the abstract, can be confusing. Perhaps this confusion is caused by our misplaced choice of the word "predictor" in the following sentence:

"Remarkably, of all the types considered, we find that ethnic diversity is **the strongest predictor of** a field's scientific impact (r is 0.77 and 0.55 for group and individual ethnic diversity, respectively)."

With this wording, the reader maybe misinterpret the subsequent r as referring to some *regression* analysis (in which case the use of R^2 is more appropriate), when in reality we were referring to *Pearson's r* in our *correlation* analysis. To rectify this, we now reword the sentence as follows:

"Remarkably, of all the types considered, we find that ethnic diversity **is the most strongly correlated with** a field's scientific impact (r is 0.77 and 0.55 for group and individual ethnic diversity, respectively)."

Finally, note that we have now added regression analysis, and report the corresponding R^2 (see Tables S4 and S5).

~ It is hard to follow which papers were actually used in the final dataset. Page 3 lists four exclusion criteria, page 10 adds two additional criteria, and page 14 lists even more filters applied to the data. Furthermore, there is a big difference between a scientist who published one paper with 10 co-authors, and one who published 10 papers each with 1 co-author. Were there baselines for minimum number of papers an author wrote when studying a career? The supplemental should better characterize the final dataset analyzed in the work.

Thank you for pointing out this shortcoming. To make this more accessible to the reader, we have collected all of this information into a single table which has also been added to the Supplementary Materials; see Table S7 (shown below for your convenience). References to this table have also been added to the relevant points in the main article and the Supplementary Materials to improve readability.

Dataset	Filter	Set Size
Main dataset (\mathcal{D}). This is the dataset that will be used by default in all analyses unless stated otherwise.	Of all the papers in the Microsoft Academic Graph (MAG) dataset, we considered all papers from the top five journals from 3 randomly selected subfields from each of the 8 main fields.	1,045,401 papers, authored by 1,529,279 unique authors.
Dataset to measure Group Gender Diversity Index	Papers in \mathcal{D} where the gender of all authors are known with 90% certainty (using Genderize.io).	460,238 papers
Dataset to measure Group Discipline Diversity Index	Papers in \mathcal{D} where the discipline of all authors in a paper is clear and known (i.e. not interdisciplinary) was used.	568,269 papers
Dataset to measure Group Affiliation Diversity Index	Papers in \mathcal{D} where each author has exactly one affiliation	207,899 papers
Dataset to measure the Individual Diversity Indices	Scientists in \mathcal{D} with at least 10 collaborators	766,338 scientists with a total of 5,103,877 collaborators taken from 9,472,439 papers

~ How were careers re-constructed? How certain are you about the accuracy of this process?

The randomized baseline models were used as a means of indirectly reconstructing the authors' careers. By shuffling the characteristic of interest (be it ethnicity, academic age, gender, or affiliation) while controlling for all other factors, we created surrogate individuals with re-constructed careers which were otherwise identical to the actual subjects, allowing the effect of specific factors to be isolated and studied.

In addition, the updated manuscript now contains a more detailed description of this randomization process (see Section S4).

~ The description of Academic Age is confusing. Was it calculated at the year of publication or for all scientists in 2009? The text reads as the latter, which would not capture the true nature of academic age in the given context. I think this is just a writing issue, but please clarify.

Thank you for spotting this. Now, when commenting on academic age in Section S2.1, we say:

...“age” refers to the academic age of a scientist, which is measured in each paper, p , by subtracting the year of the scientist's first paper from the year in which p was published.

~ Fig 3 would be easier to follow if 2 different tones were used to differentiate real vs randomized, and then 2 shades denoted diverse vs non-diverse.

Agreed. We have now done exactly that; see the new version of Figure 3.

~ While Table 1 and 2 provide sufficient evidence to support the conclusions of the CEM, they do not illustrate the intuition behind the method to an un-experienced reader. I have a feeling this paper might attract many people who are not familiar with CEM. Please provide an illustration of the treatment v control matching process, and highlight the trend that a reader is supposed to identify.

Thank you for pointing this out. We have created an infographic which illustrates this process (please see Figure S14).

Reviewer #3:

Review for NCOMMS-18-16552: Ethnic Diversity Increases Scientific Impact

This paper analyzes millions of paper outputs, spanning 24 fields and millions of scientists to examine how different types of diversity (ethnicity, disciplinary, gender, affiliation, and age) correlate with scientific impact. Distinctively, the authors generate novel coding of scientist ethnicities, ages, and gender, and construct (new) diversity measures across both the group and individual level. Overall, this paper's focus on the compositional determinants of scientific performance across a broad swath of academia is welcome and of interest to the Nature Communications readership. However, I found significant concerns with this paper. I outline these concerns below, and hope that you find them helpful.

1. Most importantly, as the authors note, patterns that they present are correlations, not causal. As a result, how meaningful and useful are these univariate correlations? I believe your results of a positive link between ethnic diversity and citation counts. Nonetheless, there are many possible explanations for this link. Off the top of my head, more productive scientists run more international labs, as they reputation allows them to draw across populations. Their prominence could also allow these "stars" to coauthor more broadly. In other words, the correlation could go both ways: scientific impact could result in more ethnically diverse groups. At this stage, it's hard to unpack what is actually going on.

Where possible we have attempted various ways of accounting for alternative explanations and confounding factors, including: number of authors, year of publication, previous author success, academic age, number of collaborators, and university ranking (see Section S3 for the newly-added analysis of the link between university ranking and ethnic diversity). This list is by no means exhaustive, but as is always the case with observational studies, the set of factors that can be included is invariably subject to the scope and availability of the data.

Having said that, we agree that distinguishing between causation and correlation is notoriously difficult. Hence, we have reviewed the manuscript in light of your comments and have made a number of adjustments to tone down our claims and thus provide the reader with a more nuanced account of

this very complex issue. All changes are highlighted in the main text in blue, but most notably, we have also changed the title of the article to:

"Ethnic Diversity is Associated with Scientific Impact"

Instead of:

"Ethnic Diversity Increases Scientific Impact"

2. You make a number of laudable attempts to drive in the direction of causality. However, I remain unconvinced, while noting that a causal story is incredibly difficult for the type of question you are asking. The randomized comparison set is a bit of a straw-man if you think about it. If I were a skeptic, I would argue that you cherry-picked a result and the randomized sample is merely regression to the mean. For the CEM, the identifying assumption remains a comprehensive set of covariates, and this assumption is untestable.

We agree with this sentiment, and to some extent we hope that our response to the previous comment addresses your concern, at least partially. In addition, we would like to point out that in cases where direct interventional studies are not possible, matching techniques in general and CEM in particular are accepted in many contexts as one way of inferring causality (for e.g., see *Catalini et al., "The incidence and role of negative citations in science", PNAS 2015*).

3. Ultimately, I think it would benefit this paper to stick to its strengths, which is the data itself. There is no need (or expectation) that the patterns you present are causal. Leave that to the small-group experimental psychologists. Merely present this paper as an interesting fact, presented across a comprehensive window into academia. By claiming less, I think the paper would be more powerful.

Following your recommendation, we have now toned down our causal claims throughout the paper.

4. What is the actual number of publications used? Is it 9 million? 1 million? It's hard to decipher your empirics here.

Indeed, this was not clear enough. Depending on the exact analysis being conducted, the data had to be sliced and diced in different ways. However, we do agree that this process can be somewhat confusing to the reader as the descriptions of these different subsets are distributed throughout the article.

For better readability, we collected all of this information into a single table which has been added to the Supplementary Materials (see Table S7). This table is also referenced at the relevant points in the text.

5. I don't see that much a difference between individual and group diversity (e.g., Figure 4). If we don't see that many differences, is the distinction itself important?

We believe it is important because it relates to the mechanism behind the observed correlations - one metric reflects *individual* characteristics such as open-mindedness and receptiveness towards other cultures, while the other reflects *group* characteristics such as the effect of diverse perspectives and ideas.

This point was mentioned in the paper, but we now emphasize it further by re-wording the relevant sentence as follows:

"...In other words, having co-authors who are inclined to collaborate across ethnic lines (i.e., co-authors whose *individual* ethnic diversity is high) appears to be not as important as the mere presence of co-authors of different ethnicities (i.e., co-authors whose *group* ethnic diversity is high)."

6. I might play up changes in time more. Understanding the changing role of diversity would be particularly interesting.

Thank you for pointing this out - we definitely agree that this is an important aspect that needs to be studied. We have already conducted experiments which look at the effect of diversity on scientific impact (see Figures 3E - 3G).

To further explore the issue of changes in time, we have now conducted additional experiments to study the effect of time on four types of diversity (presented in Figure 1B). In all cases, the results appear to be consistent with our overall findings.

7. As you might infer, I am not a CS scholar. With all due respect, I might go ahead and run some multiple regressions, which would allow for the necessary controls to be incorporated and for more robust correlations. This also might make the results more easily translated to the social science sphere.

We agree, and have now conducted a number of multiple regressions. These are presented in Tables S4 and S5, from which it can be seen that the results were clearly consistent with our overall conclusions.

8. Hats off for collecting a great dataset. I do think this is a promising project.

Thank you for the encouragement, as well as for the many constructive suggestions and comments.

Reviewers' comments:

Reviewer #1 (Remarks to the Author):

I am fully satisfied by the revision made by the authors and the point to point answers and the supplementary material added.

I recommend the publication of the manuscript in this revised form.

Reviewer #2 (Remarks to the Author):

Overall the authors made only minor improvements to the work and many of my initial concerns remain. I am now worried these issues permeate many of the results.

Therefore, I cannot currently endorse the article's publication in Nature Communications.

General notes:

~ I am still concerned about the randomization process employed in this study, and which the authors have neglected to modify. The consequence of an inappropriate randomization are clearly shown in Fig 1B age. Note the huge difference between the real data and their randomization for early years. This occurs for the simple reason that all authors publishing in 1960 must have a low academic age. However, the randomization process used here does not account for the population composition over time, and we do not see a similar drop in the randomized data. As the authors now clarify in their SI, the only feature constraining randomization is the subfield. But as myself and the other reviewers have noted, the no of authors per paper and publication year are arguably more important features of the data which also need to be controlled at the same time.

Note that in Fig 1B age, the real data quickly approaches the randomized data for later years (where the data better reflects the population's distribution of academic ages). This suggests that there is very little evidence to support homophily for academic age (a more expected result since many publications result from advisor-advisee collaboration). I am concerned similar effects may underly the other results in this paper.

~ I agree with Reviewer 3 that the paper would benefit from a multiple regression analysis for all results in Fig 2 (again, not just ethnic diversity) and these should be presented in the main text. In particular, all three confounding variables (subfield, year, no of authors) should be considered in the same regression.

That being said, can the authors explain why there is a relatively strong negative effect of publication year on citation count? It was my understanding that the normalization by average citation count in a year should remove any relationship between the two factors.

~ There is insufficient discussion of Fig 3 in the main text. Is this randomization process the same as Fig 1?

If yes - then the major concerns from above remain; if no - then what process was used?

~ The authors provide a confusing response to my request to see distributions of co-variates in the

CEM (note also, this is the concern raised by reviewer 1). When using CEM, one should always check that the binning used by the matching process is fine-enough to equalize the distributions of co-variables in the matched samples. If the matching is too coarse, then the effect of co-variables are not removed from the samples. Are the distribution plots (Fig S15, S16) before or after matching?

If before - ok, but its not surprising that the control and treatment groups are somewhat different. Please reproduce for after.

If after - these plots raise a few concerns! The purpose of a CEM is to create 2 groups that are roughly identical on all co-variables. However, these distributions demonstrate that the groups are not sufficiently identical (the authors report very significant chi-squared values rejecting the null hypothesis of equality). For example, consider the distributions for the number of authors (Fig S15, second row). There are rather significant differences in the number of low-author papers. This co-variate has repeatedly been related to the citation count. All of these plots suggest the CEM was not sufficient to remove the confounding factors.

Note also that the above situation follows from a sensitivity to binning of covariates in the matching process, which is different from the sensitivity to percentile defining control and treatment groups addressed by the authors.

~Fig 3a,b shows that the major difference between the real and randomized data is the large peak at $d_{eth}=0$ (no ethnic diversity at all). This suggests a sufficient criteria is just some group ethnic diversity ($d_{eth} > 0$).

~ I leave it for the editor to decide, but speaking as a scholar in science of science, I would want to see a discussion of the field-dependent and time-dependent results in the main text, not only the SI. These can be address for either group or individual diversity measures, with slight differences in the interpretations. Specifically:

- + Are some disciplines significantly more homophilic than others?
- + How have all diversity indices changed over time in the disciplines? ie. Are some disciplines embracing diversity better than others?
- + This is important for all diversity types. For example, is affiliation diversity changing and does it lead to greater impact? this has important consequences since many large grants emphasize multi-site teams. maybe this change is larger in fields where the granting agencies play a bigger role (ie health and bio).

Presentation notes (again, at the discretion of the editor):

~ I'm not sure if the title accurately captures the breadth of this paper.

~ The abstract does not offer a sufficient introduction to the material, and instead focuses strongly on methods.

~ The main text is very methods heavy, with little interpretation of the results.

~ The paper would greatly benefit from a graphic illustration of group diversity and individual diversity. My original suggestion was to leverage Fig 4 for this purpose, but the addition of another figure to illustrate the difference would also suffice.

Specific notes:

The description of the dataset has been improved, but it is still missing several important details:
+ The MAG uses a proprietary name disambiguation algorithm to identify the papers belonging to each author. How might errors in this disambiguation process affect your results? (Specifically for the individual diversity indicators).

+ It appears an individual's affiliation is provided by the MAG. What affiliation does this represent? The current/last known affiliation? Or the affiliation corresponding to the majority of an author's work? An author that changes their own affiliation multiple times would have a high-affiliation diversity in the co-authorship network. Maybe the correct measure is affiliation relative to the author's affiliation at the time of publication?

Reviewer #3 (Remarks to the Author):

Review for NCOMMS-18-16552a: Ethnic Diversity is Associated with Scientific Impact

I will begin by stating that this revised manuscript has addressed my concerns and I believe that it can move forward towards publication. Nonetheless, I believe that there are minor issues, imminently doable by the authors, that will lend clarity to the article. Thus, I have one main point as follows.

To summarize, this article studies five types of diversity (e.g., ethnicity, discipline, etc) and for each of the five types, the authors study group diversity (i.e., the heterogeneity of a paper's set of authors) and individual diversity (i.e., the heterogeneity of a scientist's entire set of collaborators).

I think that this typology can be clarified. A first quibble is with the use of the word "type". Is the word you want to use instead "class"? In my opinion, and I admit I might be wrong, a type seems more fitting with how diversity is constructed (e.g., Gini, Herfindahl) and class is the dimension (e.g., age, ethnicity) along which that diversity measure is constructed.

Along these lines, I might separate out group and individual diversity, and be very clear that these are not just more types added to the list, but rather these two constructs are the "level" at which diversity measures are constructed. In other words, you have 5X2 measures of diversity and you want to be clear that the two dimensions are distinct.

Moreover, I would think hard about what to call them. Group diversity, taken alone, could mean many things. I might think about the diversity within a scientific group or lab, which is clearly not the variable the authors have constructed. Thus, I might suggest one of the following: paper diversity/paper-author diversity/project diversity that more closely aligns with your construct.

Similarly, individual diversity seems imprecise. I might simply call this "collaborator diversity". At some point, I would point out that this latter measure is an egocentric (i.e., centered around the focal author) measure.

Good job with the revision!

Dear Reviewers,

Thank you for your time and effort reviewing our work and for your comments, which helped us to greatly improve and significantly expand our manuscript. Note that these include several additional analyses, which led to a number of interesting new insights (please see the new version of the manuscript for more details).

Below, we specify our responses (highlighted in blue) to every comment.

Reviewer #1:

I am fully satisfied by the revision made by the authors and the point to point answers and the supplementary material added. I recommend the publication of the manuscript in this revised form.

Thank you for the constructive comments and for the time and effort that you have put into this.

Reviewer #2:

Overall the authors made only minor improvements to the work and many of my initial concerns remain. I am now worried these issues permeate many of the results.

Therefore, I cannot currently endorse the article's publication in Nature Communications.

General notes:

~ I am still concerned about the randomization process employed in this study, and which the authors have neglected to modify. The consequence of an inappropriate randomization are clearly shown in Fig 1B age. Note the huge difference between the real data and their randomization for early years. This occurs for the simple reason that all authors publishing in 1960 must have a low academic age. However, the randomization process used here does not account for the population composition over time, and we do not see a similar drop in the randomized data. As the authors now clarify in their SI, the only feature constraining randomization is the subfield. But as myself and the other reviewers have noted, the no of authors per paper and publication year are arguably more important features of the data which also need to be controlled at the same time.

Note that in Fig 1B age, the real data quickly approaches the randomized data for later years (where the data better reflects the population's distribution of academic ages). This suggests

that there is very little evidence to support homophily for academic age (a more expected result since many publications result from advisor-advisee collaboration). I am concerned similar effects may underly the other results in this paper.

We appreciate the reviewer's concerns, and see now that these concerns were not fully addressed in the first round of revisions. To rectify this, the randomized model has been updated so that the shuffling only occurs between authors of papers which are identical with respect to the following confounding factors: subfield, year of publication, and number of authors; see the new versions of Figures 1, 4, S4, S5, S6, and S7, as well as the highlighted modifications in Section S4, which contains a detailed description of the updated procedure.

These adjustments allow the targeted class (i.e. ethnicity, age, gender or affiliation) to be more effectively isolated, producing results that seem clearer and more consistent with common sense notions of academic collaboration. In the case of academic age, for example, not only is there no longer any measurable homophily, it appears that scientists actually seem to selectively work with co-authors of different academic ages, an outcome which agrees with the reviewer's expectations. For the other classes of diversity, homophily was observed to persist but here too the results seem to be cleaner and more intuitive. For example, in Figure 1B (affiliation), there is a large increase in the 1990s, which is contemporaneous with the emergence and rapid growth of the internet; see the new discussion highlighted in blue in page 3 of the main manuscript.

~ I agree with Reviewer 3 that the paper would benefit from a multiple regression analysis for all results in Fig 2 (again, not just ethnic diversity) and these should be presented in the main text. In particular, all three confounding variables (subfield, year, no of authors) should be considered in the same regression.

We agree that this would improve the regression tables. Recall that, in the previous version of the manuscript, we had two regression tables (both concerning ethnic diversity): one table included publication year and subfield; the other included publication year and no. of authors. These two have now been replaced with a single regression table which includes all three confounding factors; our findings regarding ethnic diversity still hold. We also created five versions of this table; one for every class of diversity. We now include all tables below, as well as in the main text.

(A) Group ethnic diversity

	Citation Count, c_5^G							
	Engineering & Computer Science	Health & Medical Sciences	Business, Economics & Management	Humanities, Literature & Arts	Physics & Mathematics	Social Sciences	Chemical & Material Sciences	Life Sciences & Earth Sciences
d_{eth}^G	7.40*** (2.44)	3.00*** (0.64)	5.21*** (1.64)	4.77*** (1.79)	8.04** (3.30)	4.39** (1.89)	4.29** (1.95)	3.94*** (1.45)
University Ranking	-1.22*** (0.39)	-1.08*** (0.08)	-0.60** (0.24)	-0.52** (0.26)	-0.16 (0.46)	-0.55* (0.29)	-0.35 (0.29)	-1.35*** (0.23)
Author's Prior Impact	0.62*** (0.01)	1.24*** (0.01)	1.52*** (0.01)	1.61*** (0.01)	0.72*** (0.02)	1.51*** (0.01)	1.60*** (0.01)	1.53*** (0.01)
Year of Publication	0.20 (0.21)	0.24*** (0.01)	0.07 (0.10)	0.48*** (0.10)	0.13 (0.16)	0.37** (0.17)	0.24 (0.17)	0.24*** (0.01)
Number of Authors	0.00 (0.27)	0.59*** (0.15)	0.23 (0.17)	0.27 (0.18)	1.06 (1.03)	0.46** (0.19)	0.51*** (0.19)	0.69*** (0.11)
const	2221.02*** (270.36)	598.55*** (22.94)	1081.71*** (114.27)	1085.84*** (124.13)	1289.91*** (230.16)	2142.17*** (194.14)	1813.42*** (188.89)	2750.75*** (144.35)
R^2	0.11	0.24	0.33	0.35	0.19	0.34	0.35	0.39
N	139705	288827	38938	47141	146574	158479	88300	137437

Standard errors in parentheses. * $p < .1$, ** $p < .05$, *** $p < .01$ **(B) Group age diversity**

	Citation Count, c_5^G							
	Engineering & Computer Science	Health & Medical Sciences	Business, Economics & Management	Humanities, Literature & Arts	Physics & Mathematics	Social Sciences	Chemical & Material Sciences	Life Sciences & Earth Sciences
d_{age}^G	0.59 (3.41)	8.45*** (0.71)	15.06*** (1.52)	19.82*** (2.73)	10.92*** (3.37)	23.23*** (3.38)	11.41*** (2.44)	11.28*** (1.95)
University Ranking	-1.41*** (0.39)	-1.04*** (0.08)	-0.60** (0.24)	-0.51** (0.26)	-0.10 (0.46)	-0.55* (0.29)	-0.34 (0.30)	-1.31*** (0.23)
Author's Prior Impact	0.62*** (0.01)	1.24*** (0.01)	1.52*** (0.01)	1.61*** (0.01)	0.72*** (0.02)	1.51*** (0.01)	1.60*** (0.01)	1.53*** (0.01)
Year of Publication	0.22 (0.21)	0.28*** (0.01)	0.38*** (0.09)	0.04 (0.07)	0.08 (0.16)	0.42* (0.23)	0.14* (0.09)	1.09*** (0.11)
Number of Authors	0.18 (0.28)	0.24 (0.15)	0.17*** (0.06)	-0.02 (0.76)	0.74 (1.04)	0.00 (0.21)	0.63 (0.49)	0.56*** (0.12)
const	2221.02*** (270.36)	598.55*** (22.94)	1081.71*** (114.27)	1085.84*** (124.13)	1289.91*** (230.16)	2142.17*** (194.14)	1813.42*** (188.89)	2750.75*** (144.35)
R^2	0.11	0.24	0.32	0.31	0.19	0.34	0.32	0.38
N	139705	288827	38938	47141	146574	158479	88300	137437

Standard errors in parentheses. * $p < .1$, ** $p < .05$, *** $p < .01$ **(C) Group gender diversity**

	Citation Count, c_5^G							
	Engineering & Computer Science	Health & Medical Sciences	Business, Economics & Management	Humanities, Literature & Arts	Physics & Mathematics	Social Sciences	Chemical & Material Sciences	Life Sciences & Earth Sciences
d_{gen}^G	-6.34 (4.48)	-0.93 (1.38)	0.57 (1.67)	1.54 (3.38)	1.55 (4.41)	-0.24 (2.60)	6.34** (2.93)	-0.85 (2.09)
University Ranking	-0.75 (0.56)	-0.69*** (0.12)	0.06 (0.19)	-1.72*** (0.41)	-0.11 (0.59)	-0.68** (0.29)	-1.11*** (0.35)	-0.92*** (0.29)
Author's Prior Impact	1.33*** (0.02)	1.67*** (0.02)	0.92*** (0.01)	1.53*** (0.04)	0.65*** (0.03)	1.47*** (0.01)	1.06*** (0.05)	1.61*** (0.01)
Year of Publication	0.70** (0.35)	0.22*** (0.03)	0.34*** (0.10)	0.07 (0.08)	0.02 (0.21)	0.22 (0.23)	0.05 (0.10)	1.04*** (0.15)
Number of Authors	-0.13 (0.36)	0.79*** (0.19)	0.38*** (0.06)	1.44* (0.78)	1.75 (1.27)	1.12*** (0.19)	1.13** (0.51)	0.76*** (0.13)
const	946.57** (409.64)	541.77*** (41.14)	2617.85*** (104.67)	468.14*** (116.18)	1579.15*** (304.95)	2669.66*** (235.49)	784.17*** (133.25)	2787.59*** (183.41)
R^2	0.16	0.29	0.32	0.31	0.17	0.39	0.26	0.41
N	58288	188249	14904	8911	36949	30420	50887	71630

Standard errors in parentheses. * $p < .1$, ** $p < .05$, *** $p < .01$

(D) Group affiliation diversity

	Citation Count, c_{ij}^G							
	Engineering & Computer Science	Health & Medical Sciences	Business, Economics & Management	Humanities, Literature & Arts	Physics & Mathematics	Social Sciences	Chemical & Material Sciences	Life Sciences & Earth Sciences
d_{aff}^G	-2.85 (2.35)	2.93*** (0.60)	2.45** (0.97)	0.85 (2.70)	9.88*** (3.35)	5.77*** (1.97)	0.43 (2.26)	3.89*** (1.36)
University Ranking	-1.35*** (0.39)	-1.16*** (0.08)	-0.12 (0.18)	-1.29*** (0.36)	-0.26 (0.46)	-0.59** (0.30)	-0.79*** (0.30)	-1.42*** (0.24)
Author's Prior Impact	0.62*** (0.01)	1.23*** (0.01)	0.92*** (0.01)	1.49*** (0.03)	0.72*** (0.02)	1.60*** (0.01)	1.04*** (0.04)	1.53*** (0.01)
Year of Publication	0.14 (0.21)	0.25*** (0.01)	0.28*** (0.09)	0.13* (0.07)	0.10 (0.16)	0.58** (0.23)	0.06 (0.09)	1.04*** (0.11)
Number of Authors	0.26 (0.28)	0.55*** (0.15)	0.35*** (0.06)	1.59** (0.77)	0.71 (1.05)	0.31 (0.21)	1.24** (0.49)	0.64*** (0.12)
const	2240.33*** (275.40)	622.76*** (23.59)	2370.64*** (91.50)	327.82*** (97.70)	1336.28*** (230.77)	2319.30*** (231.07)	793.64*** (117.89)	2721.82*** (144.24)
R^2	0.11	0.24	0.32	0.30	0.20	0.35	0.25	0.39
N	38236	35925	4736	2738	61898	6431	25656	32279

Standard errors in parentheses. * $p < .1$, ** $p < .05$, *** $p < .01$

(E) Group discipline diversity

	Citation Count, c_{ij}^G							
	Engineering & Computer Science	Health & Medical Sciences	Business, Economics & Management	Humanities, Literature & Arts	Physics & Mathematics	Social Sciences	Chemical & Material Sciences	Life Sciences & Earth Sciences
d_{dep}^G	7.39 (9.91)	15.08*** (1.66)	6.92 (5.47)	31.35*** (6.68)	24.35*** (7.37)	7.00 (13.70)	25.05*** (7.08)	15.77*** (3.42)
University Ranking	-2.46*** (0.55)	-1.01*** (0.10)	-0.49 (0.30)	-1.36*** (0.51)	-0.96 (0.64)	0.28 (0.53)	-0.85* (0.48)	-1.75*** (0.32)
Author's Prior Impact	0.62*** (0.01)	1.35*** (0.01)	0.91*** (0.01)	1.45*** (0.04)	0.69*** (0.03)	1.80*** (0.02)	0.96*** (0.05)	1.55*** (0.01)
Year of Publication	0.15 (0.22)	0.28*** (0.02)	0.29*** (0.09)	0.01 (0.08)	-0.01 (0.18)	0.71*** (0.25)	0.19** (0.10)	1.13*** (0.11)
Number of Authors	0.02 (0.02)	0.05*** (0.02)	0.02*** (0.01)	0.24* (0.14)	0.28 (0.20)	0.10*** (0.03)	0.17*** (0.05)	0.04*** (0.01)
const	-253.60 (446.69)	566.42*** (32.93)	598.47*** (182.43)	24.34 (161.50)	76.50 (352.32)	-1412.96*** (502.55)	387.01** (190.31)	2278.47*** (226.61)
R^2	0.10	0.25	0.26	0.29	0.18	0.35	0.21	0.38
N	104088	141917	20801	12238	100839	24773	65607	98006

Standard errors in parentheses. * $p < .1$, ** $p < .05$, *** $p < .01$

That being said, can the authors explain why there is a relatively strong negative effect of publication year on citation count? It was my understanding that the normalization by average citation count in a year should remove any relationship between the two factors.

Thank you for pointing this out. We checked the results and realized that this was due to an omission in the pre-processing workflow used when generating this table. Due to significant missing data problems, we only include papers published after 1958, as continuous records are only available up until this point - in all of the preceding years we encountered multiple fields which had no publications at all. However, this step was omitted when preparing the original table which introduced many outliers and noisy data points for the years prior to 1958. This has now been rectified and there are no longer any significant negative correlations; see Table 1. We also added this pre-processing step to the relevant parts in the supplementary material (Section 1.2, and Table S5).

~ There is insufficient discussion of Fig 3 in the main text. Is this randomization process the same as Fig 1?

If yes - then the major concerns from above remain; if no - then what process was used?

The same randomization process is used in both Fig. 1 and Fig. 4, previously known as Fig. 3. This is now explicitly stated in the description of Fig. 4.. Note that Fig. 4 has been updated to reflect the improved shuffling process, which constrains the following three factors: subfield, number of authors, and publication year. This improved shuffling procedure was instituted in response to the first comment, and is described in the corresponding response above, as well as in Section S4.

Most notably, the citation gap between diverse and non-diverse is now negligible in the randomized data (see Figures 4F, 4H, and 4J), which really brings out the contrast between this gap and the corresponding citation gap in the real dataset. Note that this difference was already present in the previous iteration of these results, but with the latest refinement to the model, this difference is now so clear that we felt there is no longer a need for a single subfigure which compares the real and randomized datasets. With this in mind, we felt it is safe to:

- Keep every plot comparing the impact of the diverse vs. non-diverse in real data
- Keep every plot comparing the impact of the diverse vs. non-diverse in randomized data
- Omit every plot comparing real vs. randomized data in terms of the performance gap between diverse and non-diverse (since this performance gap is now clearly negligible in the randomized data).

Finally, we note that the discussion of Figure 4 has now been expanded as requested by the reviewer.

~ The authors provide a confusing response to my request to see distributions of co-variates in the CEM (note also, this is the concern raised by reviewer 1). When using CEM, one should always check that the binning used by the matching process is fine-enough to equalize the distributions of co-variates in the matched samples. If the matching is too coarse, then the effect of co-variates are not removed from the samples. Are the distribution plots (Fig S15, S16) before or after matching?

If before - ok, but its not surprising that the control and treatment groups are somewhat different. Please reproduce for after.

If after - these plots raise a few concerns! The purpose of a CEM is to create 2 groups that are roughly identical on all co-variates. However, these distributions demonstrate that the groups are not sufficiently identical (the authors report very significant chi-squared values rejecting the null hypothesis of equality). For example, consider the distributions for the number of authors (Fig S15, second row). There are rather significant differences in the number of low-author papers. This co-variate has repeatedly been related to the citation count. All of these plots suggest the CEM was not sufficient to remove the confounding factors.

Note also that the above situation follows from a sensitivity to binning of covariates in the matching process, which is different from the sensitivity to percentile defining control and treatment groups addressed by the authors.

In response to you other questions, we checked the distribution of the co-variables within the matched bins to ensure that they are reasonably equalized. In some cases, we noticed that the differences between the means were a little high. To address this issue, the bin widths were reduced to improve the match, though none of these adjustments changed our findings (recall that our findings were found to be robust to binning decisions). The table below shows the mean values of the co-variables in the corresponding bins. Note that, of the five co-variables controlled for, exact values are used for two (namely, the number of authors, and the field of study), and in these cases the values of the co-variables in the samples would hence be identical:

	Control : $d_{eth} \leq P_{10}(d_{eth})$ Treatment : $d_{eth} > P_{90}(d_{eth})$	Control : $d_{eth} \leq P_{50}(d_{eth})$ Treatment : $d_{eth} > P_{50}(d_{eth})$
Year of Publication:		
< 1990	mean(C) = 1978.8 mean(T) = 1982.1	mean(C) = 1979.0 mean(T) = 1981.1
1990 - 1994	mean(C) = 1993.3 mean(T) = 1993.9	mean(C) = 1993.3 mean(T) = 1993.6
1995 - 1999	mean(C) = 1998.2 mean(T) = 1998.4	mean(C) = 1998.2 mean(T) = 1998.3
2000 - 2004	mean(C) = 2003.2 mean(T) = 2003.3	mean(C) = 2003.2 mean(T) = 2003.2
2005 - 2009	mean(C) = 2008.2 mean(T) = 2008.3	mean(C) = 2008.2 mean(T) = 2008.2
Author's Impact:		
Bottom 25%	mean(C) = 1.6 citations mean(T) = 1.9 citations	mean(C) = 1.6 citations mean(T) = 2.0 citations
Middle 50%	mean(C) = 9.3 citations mean(T) = 9.5 citations	mean(C) = 9.3 citations mean(T) = 9.6 citations
Top 25%	mean(C) = 43.0 citations mean(T) = 48.4 citations	mean(C) = 42.9 citations mean(T) = 48.5 citations
University Ranking:		
1-100	mean(C) = 38.1 ranking mean(T) = 33.0 ranking	mean(C) = 36.3 ranking mean(T) = 33.9 ranking
Other bins	Unlike the bin "1-100", which is continuous, the other bins are categorical (this is how the rankings are provided by the 2017 Academic Ranking of World Universities", or the "Shanghai ranking"), and therefore no mean can be calculated.	

Importantly, in response to your other concern, the distribution plots mentioned (Figs. S16 and S17) are before matching was performed (we now state this explicitly in the figure captions, and in Section S5).

As requested, we show the “after” distributions for all possible archetypes below. As can be seen, the weighting process very effectively equalizes the distributions of the co-variates.

- (1) Plot 1 for the case of top and bottom 10%. Each point on the x-axis is one unique archetype (i.e. combination of bins over the co-variates).

- (2) Plot 2 for the case of top and bottom halves.

~Fig 3a,b shows that the major difference between the real and randomized data is the large peak at $d_{eth}=0$ (no ethnic diversity at all). This suggests a sufficient criteria is just some group ethnic diversity ($d_{eth} > 0$).

This is a great point. We certainly agree that it is not possible to tell from the current results whether it is sufficient to just have $d_{eth} > 0$. To verify whether this is the case, we replicated the analysis in Figures 4E, 4G and 4I (which compare the diverse vs. the non-diverse in terms of academic impact) but after excluding all papers and scientists for which $d_{eth} = 0$. The results can be found below and in Figure S14. As can be seen, even after excluding those papers and scientists, the diverse almost always outperform the non-diverse, regardless of publication year, number of authors per paper, and number of collaborators per scientist. This discussion has now been added to the main paper.

~ I leave it for the editor to decide, but speaking as a scholar in science of science, I would want to see a discussion of the field-dependent and time-dependent results in the main text, not only the SI. These can be address for either group or individual diversity measures, with slight differences in the interpretations. Specifically:

+ Are some disciplines significantly more homophilic than others?

+ How have all diversity indices changed over time in the disciplines? ie. Are some disciplines embracing diversity better than others?

+ This is important for all diversity types. For example, is affiliation diversity changing and does it lead to greater impact? this has important consequences since many large grants emphasize multi-site teams. maybe this change is larger in fields where the granting agencies play a bigger role (ie health and bio).

Thank you for these suggestions. However, respectfully, we do feel that studying the differences between the prevalence and effect of diversity in the different disciplines is clearly a hugely complex topic and we feel that fully doing it justice could be well beyond the scope of the present paper (and indeed, one can always think of other angles, such as ethnicity specific effects). We hope you agree that the scope of the paper is already extremely broad. Our fear is that any further expansion could detract from the overall cohesiveness and focus.

Hopefully, our humble perspective on this can be considered, but if it is felt that we should still include some discussion of this topic in the present paper, we present below some of our preliminary thoughts in this direction (including some new results):

Regarding the differences in homophily across disciplines:

Figures S4 to S7 study the occurrence of homophily in the various disciplines. As can be seen, differing levels of homophily were observed in the different disciplines, and for the different attributes. The disciplines with the highest level of homophily were *Nanotechnology* (ethnicity), *Cardio* (academic age), *Cardio* (gender) and *Fluid Mechanics* (affiliation).

However, no one discipline seemed to dominate. Indeed, disciplines that were highly homophilic with regards to one attribute were often significantly less so for other attributes (for example, Nanotechnology, which had the highest level of homophily for ethnicity, was amongst the lowest in academic age). This suggests that the benefits of diversity were relatively broad, and appear to transcend disciplinary boundaries.

We also studied the evolution of homophily over time in each of the 8 main fields (these new results are shown below, and can added to SI if needed), and we can add to the main manuscript the following paragraph, commenting on these new results:

On one hand, we note that the overall patterns in Fig. 1B are preserved; i.e., for ethnicity, gender and affiliation, homophily was observed in almost all fields,

while for age the reverse seems to be true. This result lends further support to the main findings of the paper. However, on the other hand, there is also significant variation between the degree of homophily observed in the different fields. Speaking very broadly, it appeared that homophily (or “inverse” homophily, in the case of age) was less pronounced in the case of (i) Business, Economics and Management, (ii) Humanity, Literature and Arts, and (iii) Social Sciences---and noticeably more pronounced in the case of the other fields.

Presentation notes (again, at the discretion of the editor):

~ I'm not sure if the title accurately captures the breadth of this paper.

We agree. We have updated the title to the following:

“Diversity, Homophily, and Scientific Impact - the Preeminence of Ethnicity over Age, Gender, Discipline, and Affiliation”

~ The abstract does not offer a sufficient introduction to the material, and instead focuses strongly on methods.

We have now updated the abstract, and made it much shorter, following the 150-word limit of abstracts in Nature Communications articles. In so doing, we omitted many technical details, and kept only the parts that serve both as a general introduction to the topic and as a brief, non-technical summary of the main results and their implications.

~ The main text is very methods heavy, with little interpretation of the results.

We have now added several paragraphs and sentences throughout the manuscript (highlighted in blue), which elaborate on the interpretation of the results.

~ The paper would greatly benefit from a graphic illustration of group diversity and individual diversity. My original suggestion was to leverage Fig 4 for this purpose, but the addition of another figure to illustrate the difference would also suffice.

We have now added a new illustration (See Figure 3), and refer to it when introducing individual diversity in the main text.

Specific notes:

The description of the dataset has been improved, but it is still missing several important details:
+ The MAG uses a proprietary name disambiguation algorithm to identify the papers belonging to each author. How might errors in this disambiguation process affect your results? (Specifically for the individual diversity indicators).

This is an open question, and resolving it satisfactorily is very challenging. Nevertheless, to obtain a general idea of how the individual diversity scores were affected by name “ambiguation”, we compared the actual diversity scores with the diversity scores calculated using a modified version of the dataset in which name ambiguation was simulated using the following procedure:

1. Each author's name is mapped to a "reduced name" - which is a combination of his/her first initial and surname (so, for e.g., "Bedoor AlShebli" becomes *B AlShebli*, "Talat Rahwan" becomes *T Rahwan*, and so on) - this process resulted in many ambiguous author namings.
2. A random selection was drawn from the pool of all authors for which ambiguity was observed. Each selected author is randomly paired with another author with the same reduced name.
3. For each pair of authors, a random paper is transferred from one author to the other.

This model simulates one commonly encountered version of how the name ambiguity process occurs: when using services such as *Google Scholar*, etc, scientists often find that papers authored by other individuals (with similar names) have been listed under their profile.

This was done for a subset of the data consisting of 41970 papers and 63274 authors who were in the *Mechanical Engineering* field. The "ambiguation" process above was applied to approximately 10% of the authors in the pool, and we studied the effects of this on ethnic and affiliation individual diversities.

We found that, in the case of d_{eth}^I , Pearson's R between the original and "ambiguated" diversity values was **0.98**, while for d_{affil}^I , it was **0.94**. In addition, to visualize the effect of the ambiguation process, we plotted the CDFs of the ambiguated and original diversities, as well as scatter plots of the two - these are shown below.

Our primary objective in the above experiment was to gain a feel for the impact of name ambiguity on the diversity estimates - and we believe that these results (particularly the R-values of 0.98 and 0.94) indicate that they were unlikely to substantially affect the overall findings of this study.

Nevertheless, these results are only preliminary, and an in-depth study of the name disambiguation problem is beyond the scope of this paper. However, at the same time, we note that this has not prevented highly insightful and influential studies (which are also affected by the name disambiguation problem) from being published in the literature, e.g.:

- Newman, Mark EJ. "*The structure of scientific collaboration networks.*" PNAS, 2001. (4751 citations)
- Barabási, Albert-Laszlo, et al. "*Evolution of the social network of scientific collaborations.*" *Physica A: Statistical mechanics and its applications*, 2002. (2599 citations)
- Moody, J., "*The structure of a social science collaboration network: Disciplinary cohesion from 1963 to 1999.*" *American Sociological Review*, 2004. (880 citations)
- Martin, T., et al., "*Coauthorship and citation patterns in the Physical Review.*" *Physical Review E*, 2013. (62 citations)

+ It appears an individual's affiliation is provided by the MAG. What affiliation does this represent? The current/last known affiliation? Or the affiliation corresponding to the majority of an author's work? An author that changes their own affiliation multiple times would have a high-affiliation diversity in the co-authorship network. Maybe the correct measure is affiliation relative to the author's affiliation at the time of publication?

Where available, MAG provides the affiliation of every author on every paper. This is what we use this when calculating diversity. Indeed, this point was not explicitly stated in our manuscript. This has now been rectified; see Section.S2.1.

Reviewer #3:

Review for NCOMMS-18-16552a: Ethnic Diversity is Associated with Scientific Impact

I will begin by stating that this revised manuscript has addressed my concerns and I believe that it can move forward towards publication. Nonetheless, I believe that there are minor issues, imminently doable by the authors, that will lend clarity to the article. Thus, I have one main point as follows.

To summarize, this article studies five types of diversity (e.g., ethnicity, discipline, etc) and for each of the five types, the authors study group diversity (i.e., the heterogeneity of a paper's set of authors) and individual diversity (i.e., the heterogeneity of a scientist's entire set of collaborators).

I think that this typology can be clarified. A first quibble is with the use of the word “type”. Is the word you want to use instead “class”? In my opinion, and I admit I might be wrong, a type seems more fitting with how diversity is constructed (e.g., Gini, Herfindahl) and class is the dimension (e.g., age, ethnicity) along which that diversity measure is constructed.

Along these lines, I might separate out group and individual diversity, and be very clear that these are not just more types added to the list, but rather these two constructs are the “level” at which diversity measures are constructed. In other words, you have 5X2 measures of diversity and you want to be clear that the two dimensions are distinct.

Following your recommendation, we now use the term “class” instead of “type” when referring to: age, gender, affiliation, discipline, and ethnicity.

Moreover, I would think hard about what to call them. Group diversity, taken alone, could mean many things. I might think about the diversity within a scientific group or lab, which is clearly not the variable the authors have constructed. Thus, I might suggest one of the following: paper diversity/paper-author diversity/project diversity that more closely aligns with your construct.

Similarly, individual diversity seems imprecise. I might simply call this “collaborator diversity”. At some point, I would point out that this latter measure is an egocentric (i.e., centered around the focal author) measure.

Thank you for this suggestion. While we certainly see your point, we do still feel that the use of “group” and “individual” diversity has some merits, and would respectfully like to present these for your consideration. If, after reading our arguments, it is still felt that the proposed terms are preferable, we would of course be willing to comply.

Firstly, while “paper diversity” and “collaborator diversity” are more descriptive, there seems to be a slight inconsistency where, *paper* refers to the grouping in which diversity is measured, while *collaborator* refers to the object of the diversity measure itself. This could also result in these terms being misunderstood (e.g., “paper diversity” can equally be understood as “the diversity of the papers of a given scientist”).

We also note your comment that the term “group diversity” could, for example, refer to diversity within a lab or research group. However, this could also be considered one of the strengths of this terminology. Using paper/collaborator diversity restricts the scope of our study to academia, while the suggested implications of our study are not necessarily restricted to just academic collaborations. The usage of more generic terms is a reflection of this aim.

Good job with the revision!

Thank you for the time and effort spent on the review, and for the helpful comments.

REVIEWERS' COMMENTS:

Reviewer #2 (Remarks to the Author):

The authors have made substantial improvements to their manuscript and have addressed many of the most pressing issues. The update to the randomization process, in particular, significantly strengthened their argument and revealed several new phenomena of fundamental importance to the science of science.

I can now recommend accepting this paper for publication in Nature Communications and congratulate the authors for the insights they were able to provide.